# Modeling oxygen transport in the brain: An efficient coarse-grid approach to capture perivascular gradients in the parenchyma

David Pastor-Alonso[1], Maxime Berg[1,2], Franck Boyer[3], Natalie Fomin-Thunemann[4], Michel Quintard[1], Yohan Davit[1], Sylvie Lorthois[1]*

1 Institut de Mécanique des Fluides de Toulouse (IMFT), UMR 5502, Université de Toulouse, CNRS, Toulouse, France, 2 Department of Mechanical Engineering, University College London, London, United Kingdom, 3 Institut de Mathématiques de Toulouse (IMT), UMR 5219, Université de Toulouse, CNRS, UPS, Toulouse, France, 4 Department of Biomedical Engineering, Boston University, Boston, Massachusetts, United States of America

* sylvie.lorthois@imft.fr

**Data Availability Statement:** All author-generated python code, as well as Comsol reference simulation data used for comparison are available

## Abstract

Recent progresses in intravital imaging have enabled highly-resolved measurements of periarteriolar oxygen gradients (POGs) within the brain parenchyma. POGs are increasingly used as proxies to estimate the local baseline oxygen consumption, which is a hallmark of cell activity. However, the oxygen profile around a given arteriole arises from an interplay between oxygen consumption and delivery, not only by this arteriole but also by distant capillaries. Integrating such interactions across scales while accounting for the complex architecture of the microvascular network remains a challenge from a modelling perspective. This limits our ability to interpret the experimental oxygen maps and constitutes a key bottleneck toward the inverse determination of metabolic rates of oxygen.

We revisit the problem of parenchymal oxygen transport and metabolism and introduce a simple, conservative, accurate and scalable direct numerical method going beyond canonical Krogh-type models and their associated geometrical simplifications. We focus on a two-dimensional formulation, and introduce the concepts needed to combine an operator-splitting and a Green's function approach. Oxygen concentration is decomposed into a slowly-varying contribution, discretized by Finite Volumes over a coarse cartesian grid, and a rapidly-varying contribution, approximated analytically in grid-cells surrounding each vessel.

Starting with simple test cases, we thoroughly analyze the resulting errors by comparison with highly-resolved simulations of the original transport problem, showing considerable improvement of the computational-cost/accuracy balance compared to previous work. We then demonstrate the model ability to flexibly generate synthetic data reproducing the spatial dynamics of oxygen in the brain parenchyma, with sub-grid resolution. Based on these synthetic data, we show that capillaries distant from the arteriole cannot be overlooked when interpreting POGs, thus reconciling recent measurements of POGs across cortical layers with the fundamental idea that variations of vascular density within the depth of the cortex may reveal underlying differences in neuronal organization and metabolic load.

from the Zenodo repository at doi: https://doi.org/10.5281/zenodo.8383820.

**Funding:** The research leading to these results has received funding from the European Research Council under the European Union's Seventh Framework Program (FP7/2007-2013) Consolidator Brainmicroflow, ERC grant agreement no. 615102 to SL, from the NIH (awards R21CA214299 and 1RF1NS110054 to SL) and from Agence Nationale de la Recherche (J452IMF23174 to SL). It was performed using HPC resources from CALMIP (Grant 2016P1541 to SL). YD was partly funded by the H2020 program Starting Bebop, ERC grant agreement no. 803074. The funders had no role in study design, data collection and analysis, decision to publish, or preparation of the manuscript.

**Competing interests:** The authors have declared that no competing interests exist.

## Author summary

The cerebral microvascular network is the logistics system that provides energy to brain cells at the right time and place. Blood flow and oxygen can now be observed dynamically in living rodents, which transformed our knowledge of the system and its role in ageing and disease. However, oxygen concentration at a given location is the result of a subtle balance between local cellular consumption, supply by neighboring vessels and their interconnections to distant ones. Thus, measurements are difficult to interpret without integrating this multi-scale component, which requires advanced computational models. This hinders our ability to bridge the gap between experiments in rodents and clinical applications in humans.

In this work, we focus on oxygen transport between vessels, leveraging recent advances in multi-scale modelling and their mathematical foundations. By this way, we formulate for the first time a simple, conservative, accurate and scalable computational model for cerebral oxygen across scales, that is able to integrate the spatially heterogenous distribution of vessels. We illustrate how this model, combined to imaging, will pave the way towards better estimates of oxygen consumption, a hallmark of neural activity that cannot be directly measured.

## 1 Introduction

Due to its highly specialized function, the brain is one of the organs with the highest basal energy demand. With essentially no substantial energy reserves, it is thus extremely vulnerable to sudden interruptions in oxygen and nutrients delivery by the blood, which can induce neuronal death within minutes with devastating consequences, e.g., for stroke victims [1]. It is also highly sensitive to chronic cerebral hypoperfusion, which can lead to progressive neurodegeneration and cognitive decline, not only in hypoperfusion dementia [2] but also, as increasingly accepted, in Alzheimer's disease [3–6]. However, despite its critical role in the transition between health and disease, many aspects of oxygen transport and metabolism in the brain remain poorly understood.

This motivated the development of high-resolution brain imaging techniques [7]. Together with the increased sophistication of experimental protocols, which enabled the brain of living rodents to be studied in various conditions including sleep, resting and awake states, these provide an unprecedented window on microvascular dynamics (e.g. diameters, red blood cell velocities, blood and tissue oxygenation, neural activity) [7–10]. However, due to the intrinsically heterogeneous and non-local nature of network flows [11–13], the results obtained in different conditions have been difficult to interpret. As we shall see next, this contributed to casting doubt on previously accepted ideas, including the fundamental idea that both structure and function of the brain microcirculation are subservient to cerebral metabolic demand.

With regard to brain function, the physiological role of neurovascular coupling, i.e. local surges in blood flow driven by increased neuronal activity (also referred to as functional hyperamia), has been questioned. On the one hand, even the baseline level of blood flow is indeed globally sufficient to supply oxygen to neurons with elevated levels of activity [12]. On the other hand, in the words of Drew [12], *"low-flow regions are an inescapable consequence of the architecture of the cerebral vasculature"* and *"cannot be removed by functional hyperemia"*. In fact, *"increases in blood flow—whether local or global—will serve only to move the location of the low-blood-flow regions, not eliminate them* [13]*"*.

With regard to structure, the local variations of vascular density have been believed for decades to reveal underlying differences in neuronal organization and metabolic load [14–16], as a result of cerebral angiogenesis being driven by their oxygen requirements [17–19]. Recent breakthroughs in brain-wide vascular network imaging and reconstruction in rodents, associated to scaling analyses, support this vision at the scale of the whole brain [20]. However, detailed measurements of periarteriolar oxygen profiles across cortical layers in awake mice, associated with estimates of the corresponding cerebral metabolic rate of oxygen, recently suggested that baseline oxygen consumption may decrease with cortical depth, from Layer I to Layer IV [10], in contrast to the known increase of capillary density [20, 21].

Solving these apparent contradictions requires the development of models integrating the non-local nature of microvascular blood flow [11, 13, 22], which account for the complex architecture of brain microvascular networks but simplify or neglect transport and metabolism within the tissue, with models of oxygen dynamics going beyond the geometrical oversimplifications associated to Krogh-type analytical descriptions [10, 23–28].

However, the computational cost of simulating oxygen transport and consumption in the brain parenchyma by standard numerical methods, such as finite volume or finite element methods, is prohibitive. In fact, they imply to finely mesh the extravascular tissue so as to resolve the strong oxygen concentration gradients building up in the vicinity of each vessel (e.g. [29]), not to mention the technical challenge of automatically meshing its complex three-dimensional volume. A popular alternative, specifically designed to solve oxygen transport in the microcirculation, formulates the problem using Green's functions [30–34]. The non-local nature of this formulation allows the description of concentration gradients around microvessels while circumventing the need for meshing the intricate geometry of the extravascular space. However, it generally relies on the infinite domain form of the Green's function, making difficult the application of boundary conditions at the limits of the tissue domain (e.g. periodic boundary conditions). Additionally, oxygen metabolism exhibits non-linear behavior [23, 35], which is challenging to describe using Green's function and generally requires additional meshing [30, 36]. This, coupled with the non-local formulation at the core of the approach, requires the creation of large and dense matrices that are computationally costly to invert. Therefore, solving oxygen transport whether using standard methods or the Green's function approach limits the size of the regions that can be considered and hinders the potential of such methods to be used in inverse problems, where measured spatial oxygen dynamics are used to deduce local metabolic rate constants or permeability coefficients, which requires to run the direct problem many times. In the latter case, the spatial resolution of the solver is much higher than that of the measurements, which requires averaging of the numerical results, deviating from an optimal allocation of computational resources.

Such challenges have been bypassed by introducing dual mesh techniques, where the extra-vascular domain is coarsely meshed independently of microvessel locations [37], or by simplifying the mesh structure, e.g. based on cartesian grids, to approximate the extravascular domain [38]. These approaches decrease the computational cost, but do not leverage recent progresses in other fields, where analytical solutions to similar problems (analogous form of equations with same underlying mathematical structure) could be used to capture the smallest features of the extravascular oxygen field (perivascular gradients). This would circumvent the need of mesh refinement around the sources. In geosciences (well or fractured reservoir modelling), for example, coupling models are often used where analytical functions help provide a relationship between the highly conductive slender structures (commonly modeled as 1D sources) and the 3D simulation domain [39–45]. In particular, in operator-splitting approaches [46–48], the scalar field (concentration, pressure, heat, etc.) is decomposed into a slowly varying contribution and a rapidly varying contribution. The former can be solved

numerically over a coarse cartesian mesh, while the later can be approximated analytically, thus enabling a precise estimation of exchanges at the vessel-tissue interface as well as an *a posteriori* highly-resolved reconstruction of the concentration field in each mesh cell.

The goal of the present paper is to revisit the problem of oxygen transport and metabolism in the brain parenchyma to introduce a simple, scalable and accurate numerical method for its direct resolution. By simplicity, we mean the ability to use cartesian mesh cells independent of vessel locations, thus avoiding meshing the extravascular space, as well as the ability to impose various boundary conditions at the outer limits of the computational domain. By scalability, we refer to a mathematical formulation of the problem at the core of which is a low-bandwidth linear system of equations, so that the numerical resolution can be fully and efficiently parallelized. By accuracy, we mean the ability to control the numerical errors even in the case of a coarse mesh. Here, we present the associated concepts in two dimensions (Section 2), so as to increase the readability of the mathematical developments. This also permits to exploit current commercial finite element solvers, which enable to obtain reference solutions of the initial boundary value problem. This enables to carefully study how the underlying simplifications translate into numerical errors in idealized test cases that sequentially challenge these assumptions (Section 3). We then show how this model helps understanding the recent counter-intuitive experimental results on cortical oxygenation and metabolism [10, 24, 26] (Section 4). Finally, we discuss how this novel approach compares to previous work and how it will provide the groundwork for computationally affordable oxygen transport and metabolic simulations, fully coupled with intravascular transport in large microvascular networks.

## 2 Model and methods

We first focus on the diffusive transport of oxygen in the brain parenchyma, i.e. the brain tissue except for blood vessels, denoted $\Omega_\sigma$ in Fig 1A, for which we present the general three-dimensional formulation in Section 2.1. We then restrict ourselves to a 2D configuration, where vessels are reduced to a collection of circular sources, as schematized in Fig 1B. This enables to maintain the readability of the mathematical developments, introduced from Section 2.2 onwards, without significant loss of generality. We finally consider oxygen consumption in Section 2.4.

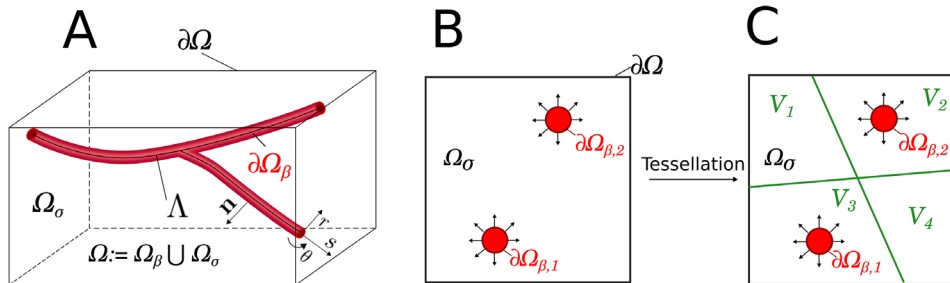

**Fig 1. Terminology and notations for parenchyma and vessel spaces.** Panel A represents a 3D region $\Omega$ of the brain tissue, which includes the parenchyma $\Omega_\sigma$ and the vessel space $\Omega_\beta$. The external boundary is denoted by $\partial\Omega$, the vessel walls by $\partial\Omega_\beta$, the vessels center-lines by $\Lambda$, the curvilinear coordinate system for the vessels by $(s, r, \theta)$ and the outer normal to the vessel walls by $\mathbf{n}$. Panel B illustrates a 2D geometry, as used in the present paper to establish the modeling framework in Section 2, with two sources. The source walls are denoted by $\partial\Omega_{\beta,1}$ and $\partial\Omega_{\beta,2}$. Panel C displays one example of a tesselation of space $\Omega_\sigma$ into 4 sub-spaces $V_k$ for $k = \{1, 2, 3, 4\}$. Here, only two sub-spaces contain sources, *i.e.*, $E(V_1) = E(V_4) = \varnothing$, $E(V_2) = \{2\}$, and $E(V_3) = \{1\}$.

## 2.1 Diffusive transport in the brain parenchyma

Following [22, 29, 49, 50], oxygen transport in the brain parenchyma is modeled through the following boundary-value problem (BVP):

$$
\begin{cases}
\nabla^2 \phi = 0 & \text{in } \Omega_\sigma & \text{(1a)} \\
-\mathbf{n} \cdot (D\boldsymbol{\nabla}\phi) = K_m(C_v|_{R_j,\theta} - \phi|_{R_j,\theta}) & \text{on } \partial\Omega_{\beta,j} \forall j \in E(\Omega) & \text{(1b)} \\
\phi = \phi_D & \text{on } \partial\Omega & \text{(1c)}
\end{cases}
$$

where spatial domains $\Omega$, $\partial\Omega$ and $\partial\Omega_{\beta,j}$ and outer normal $\mathbf{n}$ are defined in Fig 1 and $\phi$ [mol · m$^{-3}$] and $D$ [m$^2$ · s$^{-1}$] are the molar concentration field and the diffusion coefficient in the parenchyma, $C_v$ [mol · m$^{-3}$] is the intravascular molar concentration, $K_m$ [m · s$^{-1}$] is the diffusive permeability of the vessel wall and $R_j$ [$m$] the radius of vessel $j \in E(\Omega)$. $E(\Omega)$ is the set of all vessels located in the domain, so that the total number of sources ($S$) is equal to the number of vessels, i.e., to the cardinality of $E(\Omega)$ ($S$ = Card($E(\Omega)$)). In the example displayed in Fig 1A, $E(\Omega)$ = {1, 2} and $S$ = 2. To keep the developments as simple as possible, we present the model with Dirichlet boundary conditions (BCs) (Eq 1c), but our approach is readily available using Neumann and Periodic BCs as shown in the Results Section. Moreover, we follow [29, 36] and formulate the problem in terms of molar concentration, while most authors in the field use oxygen partial pressures [30, 37, 51, 52]. Partial pressures are indeed only strictly defined for a gas in a mixture of gases. The concept of partial pressure of oxygen in blood implicitly refers to gas-liquid equilibrium and can be manipulated in the case of a system at constant temperature and total pressure. Thus, we prefer to adopt in this paper a more general description of a multicomponent liquid mixture based on concentrations, as illustrated for instance in [53]. This can be accurately applied to any thermodynamic conditions, and offers a more versatile description of gas-liquid equilibrium, avoiding problems when, for instance in free-diving or high altitude, Henry's law coefficient is pressure dependent.

Due to the large aspect ratio of vessels and their low density in the tissue space, we neglect the azimuthal variations of the concentration field around the vessel walls so that Eq 1b simplifies to:

$$
-\mathbf{n} \cdot (D\boldsymbol{\nabla}\phi) = \frac{q_j(s)}{2\pi R_j} \quad \text{on } \partial\Omega_{\beta,j} \tag{2}
$$

where $q_j(s)$ [mol · m$^{-1}$·s$^{-1}$] is the integral molecular flux per unit length through the vessel wall at curvilinear abscissa $s$, defined as [22]:

$$
q_j(s) = K_{eff}(\langle C_v(s)\rangle_j - \overline{\phi}_j(s)) \tag{3}
$$

Here, $\langle C_v(s)\rangle$ is the cross-section averaged intravascular concentration:

$$
\langle C_v(s)\rangle_j = \frac{1}{\pi R_j^2} \iint_{\Omega_{\beta,j}} C_v(s, r, \theta) dS \tag{4}
$$

$\overline{\phi}_j$ is the perimeter-averaged extravascular concentration:

$$
\overline{\phi}_j(s) = \frac{1}{2\pi R_j} \int_{\partial\Omega_{\beta,j}} \phi(s, R_j, \theta) dl \tag{5}
$$

Finally, $K_{eff}$ [m$^2$ · s$^{-1}$] can be deduced from the adimensional effective reaction rate that accounts for the impact of intravascular concentration gradients on the overall flux at the

vessel wall: $K_{eff} = 8\pi \frac{D_\beta}{1+\frac{4D_\beta}{K_m R_j}}$, where $D_\beta$ is the diffusion coefficient in blood and $K_m$ [m · s$^{-1}$] is

the diffusive permeability of the vessel wall, as established for weak vessel-tissue couplings in [22]. This enables the use of the cross-section average intravascular concentration in Eq 3.

Thus, the previous BVP simplifies into:

$$\begin{cases} \nabla^2 \phi = 0 & \text{in } \Omega_\sigma & \text{(6a)} \\ -\mathbf{n} \cdot (D\boldsymbol{\nabla}\phi) = \dfrac{q_j(s)}{2\pi R_j} & \text{on } \partial\Omega_\beta & \text{(6b)} \\ \phi = \phi_D & \text{on } \partial\Omega & \text{(6c)} \end{cases}$$

together with Eqs 3–5 which are needed to estimate $q_j(s)$ in Eq 6b. Of course, a transport model in the intravascular network [22] is also needed to define $C_v(s)$, hence $q_j$, so that the developments in the present work focus on transport in the parenchyma and its coupling with the embedded intravascular network.

From now on, we restrict ourselves to a 2D configuration so that we can eliminate $s$ from Eqs 3–5 and 6b. As we shall see in Section 5, the 2D problem allows us to focus on radial transport, which provides the high perivascular concentration gradients and therefore poses the greatest challenge for the development of numerical approaches.

## 2.2 Operator-splitting

Getting inspiration from a large body of literature about mixed-dimensional problems, from well or fractured reservoir modelling in geosciences [39, 41, 54, 55] to multi-scale finite volume or operator-splitting methods in applied mathematics [45, 48, 56–58], we rewrite the previous BVP (Eq 6) by decomposing the concentration field into a slowly varying contribution $s$ and a rapidly varying contribution $r$:

$$\phi(\mathbf{x}) = s(\mathbf{x}) + r(\mathbf{x}) \tag{7}$$

so that $r$ will account for the large near-source concentration gradients while $s$ will account for the slower contributions of the domain boundary and the sources located further away.

We further introduce a tesselation $\mathscr{F}$ of space $\Omega_\sigma$ into $F$ sub-spaces $V_k$, so that $\Omega_\sigma := \bigcup_{k \in \mathscr{F}} V_k$ as schematized Fig 1C. The rationale for this will be apparent in Section 2.3.2, where we present the specific analytical expression chosen for the rapid term, with a localization strategy that maintains conformity with the finite volume (FV) mesh introduced to discretize the equations in Section 2.3.

For now, let us decompose $r$ and $s$ as sums of functions which must be continuous-by-part on tesselation $\mathscr{F}$:

$$\begin{cases} r(\mathbf{x}) = \sum_{k \in \mathscr{F}} r_k(\mathbf{x}) & \text{with} \quad r_k(\mathbf{x}) = 0 \;\; \forall \mathbf{x} \notin V_k & \text{(8a)} \\ s(\mathbf{x}) = \sum_{k \in \mathscr{F}} s_k(\mathbf{x}) & \text{with} \quad s_k(\mathbf{x}) = 0 \;\; \forall \mathbf{x} \notin V_k & \text{(8b)} \end{cases}$$

and let us define $r_k$ as any function that satisfies:

$$\begin{cases} \nabla^2 r_k = 0 & \text{in } V_k & \text{(9a)} \\ -\mathbf{n} \cdot (D\boldsymbol{\nabla} r_k) = \dfrac{q_j}{2\pi R_j} & \text{on } \partial\Omega_{\beta,j} \; \forall j \in E(V_k) & \text{(9b)} \end{cases}$$

where $E(V_k)$ is the set of sources located inside $V_k$. This general definition ensures that $r_k$

accounts a minima for the rapid contribution of all sources within $V_k$. This, in turn, ensures the regularity of $\jmath_k$ within $V_k$.

Substituting Eqs 7–9 in Eq 6 yields a BVP for each $\jmath_k$:

$$
\begin{cases}
\nabla^2 \jmath_k = 0 & \text{in } V_k & (10a) \\
\mathbf{n} \cdot \boldsymbol{\nabla} \jmath_k = 0 & \text{on } \partial\Omega_\beta & (10b) \\
\jmath_k = \phi_D - \mathscr{r}_k & \text{on } \partial\Omega & (10c)
\end{cases}
$$

These BVPs will be at the basis for the numerical finite-volume resolution of $\jmath$ on a coarse mesh in the next Section. For that purpose, we also need to close the problem 10 by imposing continuity of concentrations $\phi$ and fluxes at the interfaces between any contiguous sub-spaces $V_k$ and $V_m$ of $\mathscr{F}$:

$$
\begin{cases}
\mathbf{n} \cdot (\boldsymbol{\nabla}\phi)|_{\partial V_{k,m}} = \mathbf{n} \cdot (\boldsymbol{\nabla}\phi)|_{\partial V_{m,k}} & (11a) \\
\phi|_{\partial V_{k,m}} = \phi|_{\partial V_{m,k}} & (11b)
\end{cases}
$$

where $\partial V_{k,m} = \partial V_{m,k}$ is the interface between these contiguous sub-spaces.

Using Eqs 7 and 8 to substitute for $\phi$, and reorganizing, we obtain:

$$
\begin{cases}
\mathbf{n} \cdot (\boldsymbol{\nabla} \jmath_k - \boldsymbol{\nabla} \jmath_m)|_{\partial V_{k,m}} = \mathbf{n} \cdot (\boldsymbol{\nabla} \mathscr{r}_m - \boldsymbol{\nabla} \mathscr{r}_k)|_{\partial V_{k,m}} & (12a) \\
(\jmath_k - \jmath_m)|_{\partial V_{k,m}} = (\mathscr{r}_m - \mathscr{r}_k)|_{\partial V_{k,m}} & (12b)
\end{cases}
$$

Therefore, the final BVP for each $\jmath_k$ is:

$$
\begin{cases}
\nabla^2 \jmath_k = 0 & \text{in } V_k & (13a) \\
\mathbf{n} \cdot \boldsymbol{\nabla} \jmath_k = 0 & \text{on } \partial\Omega_\beta & (13b) \\
\jmath_k = \phi_D - \mathscr{r}_k & \text{on } \partial\Omega & (13c) \\
\mathbf{n} \cdot \boldsymbol{\nabla} \jmath_k = 0 & \text{on } \partial\Omega_\beta & (13d) \\
\mathbf{n} \cdot (\boldsymbol{\nabla} \jmath_k - \boldsymbol{\nabla} \jmath_m)|_{\partial V_{k,m}} = \mathbf{n} \cdot (\boldsymbol{\nabla} \mathscr{r}_m - \boldsymbol{\nabla} \mathscr{r}_k)|_{\partial V_{k,m}} & (13e) \\
(\jmath_k - \jmath_m)|_{\partial V_{k,m}} = (\mathscr{r}_m - \mathscr{r}_k)|_{\partial V_{k,m}} & (13f)
\end{cases}
$$

where $\mathscr{r}_k$ will be given as analytic functions of variables $q_j$ in Section 2.3.2 and $\jmath_k$ will be obtained numerically. Such a set of BVPs could typically be further discretized and solved by domain decomposition methods [56]. Here however, the strong perivascular gradients are accounted for by the rapid term. To minimize the number of unknowns, we introduce in the next Section a FV discretization where a single grid-cell is associated to each sub-space $V_k$ of tessellation $\mathscr{F}$.

## 2.3 Assembly of a system of discrete algebraic equations

For that purpose, we set tessellation $\mathscr{F}$ to match a cartesian grid of cell side-length $h = |\partial V_{k,m}|$, where $m$ is a direct neighbour of $k$ (i.e $m \in \mathcal{N}^{\prime k}$) with:

$$
\mathcal{N}^{\prime k} := \{n, s, e, w\} \tag{14}
$$

as defined in Fig 2.

From this point forward, we use symbol $\sim$ to represent the discrete average values of a field on each FV cell $k$. Noteworthy, for harmonic functions such as $\jmath(\mathbf{x})$, from Gauss's harmonic

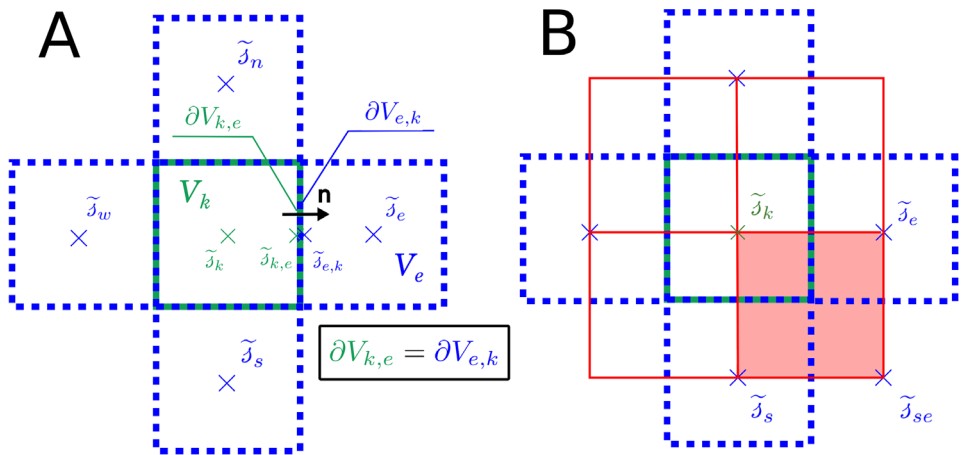

**Fig 2. Terminology and notations for the FV discretization (A) and sub-grid interpolation (B).** Panel A displays the current cell $k$ of the cartesian mesh in green and its direct neighbours $\mathcal{N}^k = \{n, s, e, w\}$ in blue. $\tilde{\jmath}_k$ is the value of the slow term at the center of cell $k$. The dummy variables $\tilde{\jmath}_{k,e}$ and $\tilde{\jmath}_{e,k}$ represent the values of the slow term on both sides of the interface $\partial V_{k,e}$. Generally, $\tilde{\jmath}_{k,e} \neq \tilde{\jmath}_{e,k}$ due to the jump introduced by Eq 12. Panel B displays the dual mesh used for sub-scale interpolation in red. This dual mesh is constructed by joining the centers of the FV grid. Its cells are denoted by numbers (0, 1, 2 and 3).

function theorem [59], if we neglect the small volume occupied by the vasculature ($\approx 3\%$ [60]), this average value can be approximated at second-order by the value at the cell's center.

Therefore, the unknowns of the system are the values of the slow term at the center of each FV cell $\tilde{\jmath}_k$, and the vessel-tissue flux for each source $q_j$. These are represented by two vectors of discrete variables $\boldsymbol{\jmath} = \{\tilde{\jmath}_1, \tilde{\jmath}_2, \tilde{\jmath}_3, ..., \tilde{\jmath}_F\}$ and $\mathbf{q} = \{q_1, q_2, q_3, ..., q_S\}$, respectively.

**2.3.1 FV discretization for the slow term.** The gradient of the slow term is approximated by the Two Point Flux Approximation (TPFA):

$$(D\boldsymbol{\nabla}\jmath_k(\mathbf{x}) \cdot \mathbf{n})|_{\partial V_{k,m}} \approx D\frac{\tilde{\jmath}_{k,m} - \tilde{\jmath}_k}{h/2} \tag{15}$$

where $h$ is the size of the FV cell face $h = |\partial V_{k,m}|$ and $\tilde{\jmath}_{k,m}$ are dummy variables, to be eliminated by substitution from the final system, which represent the values of the slow term on interfaces $\partial V_{k,m}$.

Additionally, we use the classic FV formulation by integrating Eq 13a over each FV cell $k$ (see Section A in S1 Methods for more detail). This yields:

$$-4\tilde{\jmath}_k + \sum_{m \in \mathcal{N}^k} \tilde{\jmath}_{k,m} = 0 \tag{16}$$

The discrete versions of boundary conditions 13e and 13f are:

$$\begin{cases} D\dfrac{\tilde{\jmath}_{k,m} - \tilde{\jmath}_k}{h/2} - D\dfrac{\tilde{\jmath}_m - \tilde{\jmath}_{m,k}}{h/2} = \dfrac{1}{h}\displaystyle\int_{\partial V_{k,m}} \mathbf{n} \cdot (D\boldsymbol{\nabla}r_m(\mathbf{x}) - D\boldsymbol{\nabla}r_k(\mathbf{x}))dl & \text{(17a)} \\[4mm] \tilde{\jmath}_{k,m} - \tilde{\jmath}_{m,k} = \dfrac{1}{h}\displaystyle\int_{\partial V_{k,m}} (r_m - r_k)dl & \text{(17b)} \end{cases}$$

From the above equations, we can express the dummy variables $\tilde{\jmath}_{k,m}$ as follows:

$$\tilde{\jmath}_{k,m} = \frac{\tilde{\jmath}_k + \tilde{\jmath}_m}{2} + \frac{J_{k,m}}{2} \tag{18}$$

with:

$$J_{k,m} = \frac{1}{2} \int_{\partial V_{k,m}} \mathbf{n} \cdot (\boldsymbol{\nabla}\mathscr{r}_m - \boldsymbol{\nabla}\mathscr{r}_k)dl + \frac{1}{h}\int_{\partial V_{k,m}} (\mathscr{r}_m - \mathscr{r}_k)dl \tag{19}$$

where $J_{k,m}$ is a function of the sources $\mathbf{q}$ as we shall see in Section 2.3.2, and it accounts for the discontinuities of the rapid term across the interfaces of the FV. We can express Eq 16 as a function of the unknowns of the system:

$$-4\tilde{\jmath}_k + \sum_{m\in\mathcal{N}^k} (\tilde{\jmath}_m + J_{k,m}) = 0 \tag{20}$$

where $\tilde{\jmath}_k$ and $\tilde{\jmath}_m$ are found under the vector $\boldsymbol{\jmath}$.

Moreover, if the current mesh cell $k$ belongs to a boundary, the boundary condition 10c is used instead of 17, yielding:

$$\tilde{\jmath}_{k,\partial\Omega} = \phi_D - \mathscr{r}_{k,\partial\Omega} \quad \text{if} \quad \partial V_{k,\partial\Omega} \in \partial\Omega \tag{21}$$

Therefore, the discretized version of BVP 13 can be assembled from Eqs 20 and 21 into an algebraic system with as many equations as grid-cells:

$$\mathbf{A} \cdot \boldsymbol{\jmath} + \mathbf{J} = \mathbf{b}_{\partial\Omega} \tag{22}$$

where matrix $\mathbf{A}$ contains the classic diffusion stencil, and the vector $\mathbf{J}$ contains the values of $J$ given by Eq 19. Therefore, for each row $k$, $\mathbf{A}$ contains one diagonal value and 4 off-diagonal values associated to its neighbours, while the vector $\mathbf{b}_{\partial\Omega}$ contains the entries relevant to enforce the BCs 21.

We have constructed a system of algebraic equations that enforces mass balance of the concentration field in each FV cell through Eq 20. To go further, we must specify the choice of the rapid term that will allow the the entries of $\mathbf{J}$ to be deduced from Eq 19. We note $\mathscr{r}$ could be obtained numerically as in [56, 57], or approximated analytically based on the Green's function formulation, as detailed in the next Section.

**2.3.2 Potential-based localized formulation for the rapid term $\mathscr{r}$.** We first recall that, as written in Section 2.2, $\mathscr{r}_k$ must be harmonic functions that satisfy Eq 9 for all $k$, ensuring to consider, a minima, the rapid contribution of all sources within $V_k$.

Straightforward analytical approximations for $\mathscr{r}_k$ in 2D are therefore:

$$\mathscr{r}_k = \sum_{j\in E(\widehat{V}_k)} P_j \tag{23}$$

where $\widehat{V}_k$ represents any extension of $V_k$, i.e. any region of space containing $V_k$, and $P_j$ is the single-source potential associated to source $j$.

According to potential theory [61–63], $P_j$ can be written as (see Section B in S1 Methods):

$$P_j = \begin{cases} \overline{\phi}_j + \dfrac{q_j}{2\pi D} \ln\left(\dfrac{R_j}{||\mathbf{x} - \mathbf{x}_j||}\right) & \text{if } ||\mathbf{x} - \mathbf{x}_j|| > R_j \\[3mm] \overline{\phi}_j & \text{if } ||\mathbf{x} - \mathbf{x}_j|| \leq R_j \end{cases} \tag{24}$$

With this explicit definition of the potentials, the expression of the rapid term (Eq 23) only depends on the vessel-tissue exchanges (**q**) and on the position $||\mathbf{x}-\mathbf{x}_j||$, so that $r_k = r_k(\mathbf{q}; \mathbf{x})$. From this expression of the rapid term, now we have an explicit definition of **J** from Eq 22 as a function of the vessel-tissue exchanges **q**. We can thus assemble the discrete system of equations with $\jmath$ and **q** as follows:

$$\mathbf{A} \cdot \jmath + \mathbf{B} \cdot \mathbf{q} = \mathbf{b}_{\partial\Omega} \tag{25}$$

Noteworthy, $r_k$ strictly fulfills the constraints corresponding to Eq 9 if and only if there is a single source $i$ in $\widehat{V}_k$, for which $\mathbf{x}_i \in V_k$. Any additional source $j$ in $\widehat{V}_k$ induces a perturbation $\varepsilon_{i,j}^q$:

$$\varepsilon_{i,j}^q = -\mathbf{n} \cdot (D\boldsymbol{\nabla}P_j)|_{\partial\Omega_{\beta,i}} \tag{26}$$

of the normal flux around source $i$ (see Section C in S1 Methods), which is not accounted for in the model. However, the integral contribution of these errors is null so the model remains conservative (Section C in S1 Methods). The impact of this perturbation will be examined in the Results Section 3.3.

Inspired by our previous work in [64], we set $\widehat{V}_k$ to correspond to a finite number $n^2$ of cells in the finite-volume mesh, with $n \geq 3$ to avoid a special treatment for sources lying on the interface between two mesh cells.

By this way, $n$ sets up the characteristic size of the region in which we account for the contribution of nearby sources to $r_k$, while the contribution of sources outside $\widehat{V}_k$ is only implicitly treated through $\jmath_k$, as illustrated in Fig 3. Thus, increasing $n$ leads to a better approximation of the concentration field (see Section 2.3.4), but at the same time increases the density of matrix **B** in Eq 25. In the limit case where $\widehat{V}_k = \Omega$, we would obtain an element-wise non-zero **B**, leading to a non-sparse system similar to [30, 51, 65] where the boundary integrals of the classic Green's function formulation are estimated by $\jmath$. Since the goal here is to obtain a sparse linear system, $\widehat{V}$ ($\widehat{V} \subset \Omega$) is chosen to be small in comparison to the domain of computation, but large enough to include the near source gradients.

The estimation of the single source potential based on the Green's integral formulation has a natural extension to 3D. The circular sources that appear in the 2D model provide a simple formulation to the potential since the double layer potential is null (see Section B in S1 Methods). In contrast, an open cylinder provides a non-null value for the double layer integral resulting in a second potential in Eq 24 [51] which accounts for the axial variations. The rest of the developments presented in Section 2.3, including FV discretization and localization of the slow term, can be simply extrapolated to 3D.

**2.3.3 Sub-grid reconstruction to estimate vessel-tissue exchanges (q).** The vessel-tissue exchanges are governed by Eq 3, which in 2D translates into:

$$q_j = K_{eff}(\langle C_v \rangle_j - \overline{\phi}_j) \tag{27}$$

for each source $j \in E(\Omega)$. In 3D, this equation should be coupled to an intravascular transport

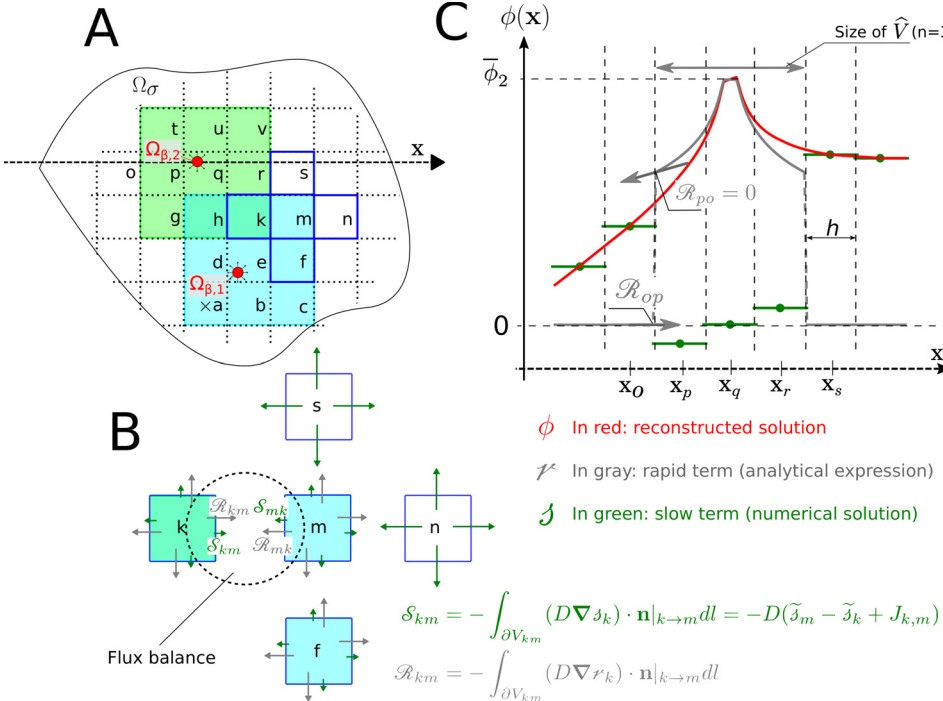

**Fig 3. Localization strategy illustrated for two sources with neighbourhood $\widehat{V}_k$ of size 3x3 grid-cells (i.e., $n = 3$).**
Panel A: Cells where the rapid term accounts for source 1 and source 2 are displayed in blue and green respectively,
with superposition in cells $h$ and $k$, so that $r_w = P_1 \, \forall w \in \{a, b, c, d, e, f, m\}$, $r_w = P_2 \, \forall w \in \{g, p, q, r, t, u, v\}$ and $r_w = P_1 + P_2 \, \forall w \in \{h, k\}$; cells lying further from sources 1 and 2 are displayed in white. In these cells, $r_w = 0$; Panel B: Flux balance for all cells highlighted in dark blue in panel A. Green arrows represent the contributions of slow terms while grey arrows those of rapid terms. The latter may exhibit jumps, e.g. at interfaces $V_{km}$ and $V_{nm}$ due to the localization-induced discontinuities in the rapid term. Panel C: Concentration field decomposition (Eq 7) along the $x$-axis crossing the center of source 2 (dashed axis in Panel A). We show in red the fine-grid reconstructed solution through Eq 30, in green the coarse-grid slow term and in grey the rapid term.

problem that introduces a discrete 1D description of average intravascular concentrations
along vessel centerlines [22] as additional unknowns (see e.g. [45, 66]). In our 2D case, how-
ever, sources are disconnected, so that the values of $\langle C_v \rangle_j$ are provided as boundary conditions.
The average wall concentration is given by:

$$\overline{\phi}_j = \frac{1}{2\pi R_j} \oint_{\partial \Omega_{\beta, j}} \phi(\mathbf{x}) dl \tag{28}$$

However, the numerical model only provides an approximation of the concentration field
at the grid-cell centers $\mathbf{x}_k$:

$$\tilde{\phi}_k = \tilde{\jmath}_k + r_k(\mathbf{q}, \mathbf{x}_k) \tag{29}$$

and, from Eq 21, at the boundary nodes.

To estimate $\overline{\phi}_j$ from Eq 28, we must reconstruct the concentration field everywhere in $\Omega_\sigma$.
For that purpose, we interpolate the slow term from its values at $x_k$ and $x_{k,\partial\Omega}$ using a classical
set of linear shape functions $\gamma_i$ associated to these points, as defined in Section D in S1 Meth-
ods. We also introduce a extended rapid term $r_i^e$ bridging the discontinuities across the inter-
faces of the FV cells $\partial V_{k,m} \forall k \in \mathcal{F} \, \& \, m \in \mathcal{N}^k$, as detailed in Section D in S1 Methods. The

resulting interpolation function $\mathscr{I}_\phi$ reads:

$$\mathscr{I}_\phi(\jmath, \mathbf{q}; \mathbf{x}) = \sum_{i \in \mathcal{T}} \gamma_i(\mathbf{x})(\tilde{\jmath}_i + \mathscr{r}_i^c(\mathbf{q}; \mathbf{x})) \tag{30}$$

where $\mathcal{T}$ represents the set of FV grid-cell centers $x_k$ and of boundary nodes $x_{k,\partial\Omega}$.

Since both $\jmath_k$ and $\mathscr{r}_k$ are harmonic functions in $V_k$, the average needed to estimate $\overline{\phi}_j$ in Eq 28 can be deduced from Gauss's harmonic function theorem, yielding $\overline{\phi}_j = \mathscr{I}_\phi(\jmath, \mathbf{q}; \mathbf{x}_j)$ so that Eq 27 becomes:

$$q_j = K_{eff}(\langle C_v \rangle_j - \mathscr{I}_\phi(\jmath, \mathbf{q}; \mathbf{x}_j)) \tag{31}$$

We can now assemble Eq 31 into a discrete linear system of $S$ equations with $\jmath$ and $\mathbf{q}$ as vectors of unknowns:

$$\mathbf{C} \cdot \jmath + \mathbf{D} \cdot \mathbf{q} = \mathbf{b}_{\partial\Omega_\beta} \tag{32}$$

Noteworthy, the interpolation function $\mathscr{I}_\phi$ uses the nearby sources as well as the four nearest FV unknowns of the mesh grid (see Section D in S1 Methods). Therefore, matrix $\mathbf{C}$ is sparse, with 4 non-zero entries per line, while the density of matrix $\mathbf{D}$ depends on the size of $\widehat{V}$. Additionally, vector $\mathbf{b}_{\partial\Omega_\beta}$ contains the values of intravascular concentrations ($\langle C_v \rangle_j$) treated here as boundary conditions.

**2.3.4 Full discrete system and error induced by localization.**   The full discrete system is therefore:

$$\begin{cases} \mathbf{A} \cdot \jmath + \mathbf{B} \cdot \mathbf{q} = \mathbf{b}_{\partial\Omega} \\ \mathbf{C} \cdot \jmath + \mathbf{D} \cdot \mathbf{q} = \mathbf{b}_{\partial\Omega_\beta} \end{cases} \tag{33}$$

with a total of $F + S$ unknowns, $F = Card(\mathscr{F})$ being the total number of FV grid-cells and $S$ the number of sources. This general form is independent of the specific choice made for the size of extensions $\widehat{V}_k$ used to define $\mathscr{r}_k$ as linear combinations of potentials based on the Green's formulation. This size, however, strongly influences the densities of matrices $\mathbf{B}$ and $\mathbf{D}$. Nevertheless, matrices $\mathbf{A}$ and $\mathbf{C}$ always remain sparse since A is the classic FV diffusion matrix with only 5 non-zero terms per line and C only depends on the interpolation function $\mathscr{I}_\phi$, resulting in 4 non-zero elements per line.

Of course, the global error resulting from approximating BVP 13 by the above system depends on the size of $\widehat{V}_k$. This global error $\varepsilon \widehat{V}$ induced by the localization strategy can be estimated by considering the neglected contribution of sources outside of $\widehat{V}_k$ to the concentration field ($\sum_{j \notin E(\widehat{V}_k)} P_j$).

The error associated to FV methods is commonly given by [44]:

$$\varepsilon_{FV} < C_0 h^2 \tag{34}$$

where $C_0$ is bounded by the norm of the second derivative of the estimated field. Using Eq 24 and considering that the minimal distance between a source in $V_k$ and one outside $\widehat{V}_k$ is of

order $(n-1)h/2$, we get an upper-bound of $\varepsilon_{\hat{V}}$:

$$\varepsilon_{\hat{V}} \leq \sum_{j \notin \hat{V}_k} \frac{4q_j}{2\pi D(n-1)^2 h^2} O(h^2) \tag{35}$$

Simplifying, we obtain:

$$\varepsilon_{\hat{V}} < \sum_{j \notin \hat{V}_k} \frac{4q_j}{2\pi D(n-1)^2} O(1) \tag{36}$$

Therefore the localization error $\varepsilon_{\hat{V}}$ is expected to decrease with $n^2$, i.e., $\varepsilon_{\hat{V}} \propto \frac{1}{(n-1)^2}$.

## 2.4 Metabolism

Now that we have introduced the concepts and formulation for the non-reactive problem (BVP 6), we introduce tissue consumption, that we model by a Michaelis-Menten reaction kinetic [67]. In the resulting reactive problem, Eq 6a is thus substituted by the following non-linear PDE:

$$D\nabla^2\phi = M\frac{\phi}{\phi+K} \quad \text{in } \Omega_\sigma \tag{37}$$

where $M$ [mol $\cdot$ m$^{-3}$ $\cdot$ s$^{-1}$] is the maximal cerebral metabolic rate of oxygen, often denoted CMRO$_{2,max}$, and $K$ [mol $\cdot$ m$^{-3}$] represents the concentration where consumption is half of its maximum, often denoted $EC_{50}$ for $O_2$ activating oxidative phosphorylation [68]. The boundary conditions on $\partial V_k$ given in Eqs 13c–13f remain unchanged. We consider $D$ and $M$ homogeneous to rewrite the PDE 13a

$$D\nabla^2\mathit{s}_k - M\left(1 - \frac{K}{K+\phi}\right) = 0 \quad \text{for } \mathbf{x} \in V_k \tag{38}$$

The new discrete system is:

$$\begin{cases} \mathbf{A} \cdot \mathit{s} + \mathbf{B} \cdot \mathbf{q} + \mathbf{S_{metab}} = \mathbf{b}_{\partial\Omega} \\ \mathbf{C} \cdot \mathit{s} + \mathbf{D} \cdot \mathbf{q} = \mathbf{b}_{\partial\Omega_\beta} \end{cases} \tag{39}$$

where $\mathbf{S_{metab}}$ is a vector containing the integral contributions of the metabolism per FV cell:

$$\mathbf{S_{metab}} = -\begin{Bmatrix} M\left(1 - \int_{V_1}\left(\frac{K}{K + \tilde{\mathit{s}}_1 + \mathit{r}_1(\mathbf{x})}\right)dV\right) \\ M\left(1 - \int_{V_2}\left(\frac{K}{K + \tilde{\mathit{s}}_2 + \mathit{r}_2(\mathbf{x})}\right)dV\right) \\ M\left(1 - \int_{V_3}\left(\frac{K}{K + \tilde{\mathit{s}}_3 + \mathit{r}_3(\mathbf{x})}\right)dV\right) \\ \vdots \\ M\left(1 - \int_{V_F}\left(\frac{K}{K + \tilde{\mathit{s}}_F + \mathit{r}_F(\mathbf{x})}\right)dV\right) \end{Bmatrix} \tag{40}$$

## 2.5 Numerical implementation

The problem is assembled and solved using an in house code written in Python. Due to the large reduction in size allowed by the multiscale model presented, the libraries scipy and numpy for solving linear problems are adequate for the 2D simulations and test cases. An extension to 3D is possible under careful consideration and optimization of the code.

The integrals in Eqs 19 and 40 are evaluated using the second order accurate Simpson's rule of integration [59]. Furthermore, the non-linear system assembled in Eq 39 is classically solved through an iterative Newton-Raphson method (see Section E in S1 Methods).

## 2.6 Summary of model assumptions

Before examining the robustness, consistency and limitations of the above model in Section 3, we recall the two main assumptions introduced in the developments:

- *Assumption 1:* We considered that the concentration field could be split into a rapid and a slow component ($r$ and $s$, respectively). In practice, we thus considered the scale of variations of $s$ to be much larger than the size of the coarse grid $h$, so that the slow field could be accurately evaluated using Eq 20. Recalling that the slow term accounts for the contribution of the domain boundaries and of the sources located outside of $\widehat{V}$ (Section 2.2), this assumption should break down in the following cases:

  - *Case 1.1*: when $h$ is not sufficiently small compared to the scale of variation driven by the boundary conditions, i.e., in simple cases, the size of the computational domain;

  - *Case 1.2*: when a source lies near the domain outer boundaries $\partial\Omega$;

  - *Case 1.3*: when the neighbourhood $\widehat{V}$ is too small to accommodate accurately for the potentials arising from nearby sources.

- *Assumption 2:* We neglected the azimuthal variations of concentration around each source ($\phi|_{\partial\Omega_j} \approx \overline{\phi}_j$). As a result of Eq 6b, we thus neglected the azimuthal variations of flux around the source's walls. This assumption is crucial to write the potential for a single source based on Eq 24. Noteworthy, in contrast to Krogh-type models [10], these azimuthal variations are neglected only locally on the source walls. We expect this assumption to break down in the following cases:

  - *Case 2.1*: when two or more sources are lying close together, that is, when the density of sources becomes locally too large;

  - *Case 2.2*: when a source lies near $\partial\Omega$.

In the next Section, we use idealized test cases of increasing complexity that help decouple the impact of these different sources of errors and clarify the associated size constraints.

## 3 Results: Error estimation

In this Section, we first consider test cases involving a single source (Section 3.1) and a single dipole, i.e., the combination of a single source and a single sink (Section 3.2). Then, in Section 3.3, we turn to multiple sources and sinks. Noteworthy, we generically designate by "source" any vessel $j$ whose concentration is greater than the local tissue concentration, i.e., for which the resulting flux $q_j$ will be positive. In the same way, we use "sink" for any vessel $j$ whose concentration is lower than the local tissue concentration, i.e., for which the resulting flux $q_j$ will

be negative. This enables "diffusional shunts" in the parenchyma, which have been evidenced experimentally between arterioles and venules [69], to be considered.

Thus, for all simulations we assign $\langle C_v \rangle_j = \phi_{max}$ to all sources and $\langle C_v \rangle_j = 0$ to all sinks, where $\phi_{max}$ represents the oxygen concentration in penetrating arterioles at the inlet of the brain cortex. We also use a diffusion coefficient $D = 2 \times 10^{-5} \mathrm{cm}^2 \cdot \mathrm{s}^{-1}$ [25, 30, 70], an effective permeability for the capillaries of $K_{eff} = 2 \times 10^{-5} \mathrm{cm}^2 \cdot \mathrm{s}^{-1}$ [29, 30] and a maximum metabolic consumption of $M = 2.4\ \mu\mathrm{mol} \cdot \mathrm{cm}^{-3} \cdot \mathrm{min}^{-1}$ which falls within physiological range (see Table 1).

Moreover, for all test cases considered in this Section, we purposely put ourselves in *Case 1.1* above by considering relatively small domains of side $L = 240\mu$m, i.e., only 50 times larger than the source/sink radii ($R = 4.8\mu$m). In doing so, we aim at providing reasonable estimates for the upper bounds of the numerical errors.

Errors are estimated by comparison with a fine mesh finite element (FE) solution of the original BVP (Eq 1 for the linear problem or Eq 37 for the non-linear problem) without any additional modeling assumptions, in the same spirit as [29]. This reference FE solution, $\phi_{ref}$, was obtained with COMSOL Multiphysics using a triangular mesh fine enough to accommodate the contours of the circular sources, to handle the azimuthal variations of the concentration field around the sources and to ensure convergence in the estimation of $q_{ref}$, obtained by integrating the normal derivative of $\phi_{ref}$ along the vessel wall.

We define the following metrics to compare our multiscale model with this reference solution. The local errors on the vessel-tissue exchanges for each source ($q_j$) are given by:

$$\varepsilon_q^j = \frac{|q_j - q_{j,ref}|}{q_{j,ref}} \tag{41}$$

and the local errors on the concentration field at the center of each grid-cell are given by:

$$\varepsilon_\phi^k = \frac{|\tilde{\phi}_k - \phi_{k,ref}|}{\phi_{k,ref}} \tag{42}$$

**Table 1. Parameter values.** Radii $R$: see Fig 7; $\phi_{max}$: oxygen concentration in penetrating arterioles at the inlet of the brain cortex; $D$: diffusion coefficient in the parenchyma; $K_{eff}$: effective diffusive permeability of the capillary walls; $\alpha$: oxygen solubility in water at atmospheric pressure; $M$: maximum metabolic rate of oxygen; $K$: concentration where consumption is half of its maximum.

| Variable | Value | Units | References |
|---|---|---|---|
| **Microvascular parameters** | | | |
| $R_{PA}$ | 20 | $\mu$m | [10] |
| $R_{cap}$ | 4.8 | $\mu$m | [10] |
| $R_{cyl}$ | 100 | $\mu$m | [10] |
| Capillary length density | [0.8, 1.2] | $\mathrm{m} \cdot \mathrm{mm}^{-3}$ | [20] |
| $\phi_{max}$ | 137 | $\mathrm{nmol} \cdot \mathrm{cm}^{-3}$ | from [9, 34, 71] |
| **Tissue transport and consumption** | | | |
| $D$ | $2 \times 10^{-5}$ | $\mathrm{cm}^2 \cdot \mathrm{s}^{-1}$ | [25, 30, 70] |
| $K_{eff}$ | $2 \times 10^{-5}$ | $\mathrm{cm}^2 \cdot \mathrm{s}^{-1}$ | [29, 30] |
| $\alpha$ | $1.39 \times 10^{-3}$ | $\mathrm{mol} \cdot \mathrm{m}^{-3} \cdot \mathrm{mmHg}^{-1}$ | [24, 28] |
| $K$ | $\sim \phi_{max}/10$ | | from [23, 30] |
| $M$ | [0, 2.4] | $\mu\mathrm{mol} \cdot \mathrm{cm}^{-3} \cdot \mathrm{min}^{-1}$ | [10, 23, 29] |

where $\tilde{\phi}_k$ is given by Eq 29. We then define the global errors as the average of the local errors:

$$\varepsilon_\phi^g = \frac{1}{F} \sum_{k \in [1,F]} \varepsilon_\phi^k \tag{43}$$

and:

$$\varepsilon_q^g = \frac{1}{S} \sum_{j \in [1,S]} \varepsilon_q^j \tag{44}$$

where where $F$ is the number of discrete grid-cells in the cartesian mesh and $S$ is the total number of sources, i.e., $S = Card(E(\Omega))$.

We consider the error on vessel-tissue exchanges ($\varepsilon_q^g$) as the main metric to assess the model's accuracy, since proper estimation of $q_j$'s relies on an accurate evaluation of the microscale dynamics and provides crucial information on oxygen exchanged between blood and tissue. The error on the concentration field serves as a secondary metric, offering valuable insights into the interactions between sources.

We compare these errors with the errors resulting from a coarse-grid FV approach without multiscale coupling, in the same spirit as [37]. Such an approach solves the simplified BVP (Eq 6 for the linear problem or Eq 37 for the non-linear problem) by approximating the average concentration on the vessel wall ($\overline{\phi}_j$) by the value of the concentration field in the nearest FV cell $k$ ($\overline{\phi}_j = \tilde{\phi}_k$ for $\Omega_{\beta,j} \in V_k$). As a result, the exchange term is given by

$$q_j = K_{eff}(\langle C_v \rangle_j - \tilde{\phi}_k) \tag{45}$$

where $k$ is the grid-cell containing source $j$. This coupling condition is not a multiscale coupling condition, as it doesn't integrate any description of the near source concentration gradients that could compensate for the scale gap with the coarse-grid for the estimation of $q_j$. At its core, it assumes a well-mixed concentration within each mesh cell, i.e., it neglects the effect of concentration gradients near sources when using a coarse grid, generating significant errors in the estimation of $q_j$ (see Figs 4–6). On the one hand, increasing mesh discretizations can solve this issue and allow to (asymptotically) recover the influence of such gradients [38], with the significant trade-off of increased computational cost. On the other hand, including a multiscale component by reconstructing analytically the local concentration near sources as done in Section 2, allows to capture the influence of the gradients whilst allowing to use a coarse-grid discretization of the tissue space. The FV solution with resolution matching that of the coarse-grid is thus useful to illustrate the interest of the multiscale coupling at the core of the developments presented in Section 2.

For the sake of comparison, the following conventions are used in all figure legends in this Section:

- Blue lines are used for the present multiscale method while red ones are used for the coarse-grid FV model.

- Continuous lines are used for the linear, non-reactive model (Eq 33) while discontinuous ones are used when metabolism is considered, i.e. reactive model (Eq 39).

- Square markers are used to display the global errors on the vessel-tissue exchanges while triangular ones are used to display errors on the concentration field.

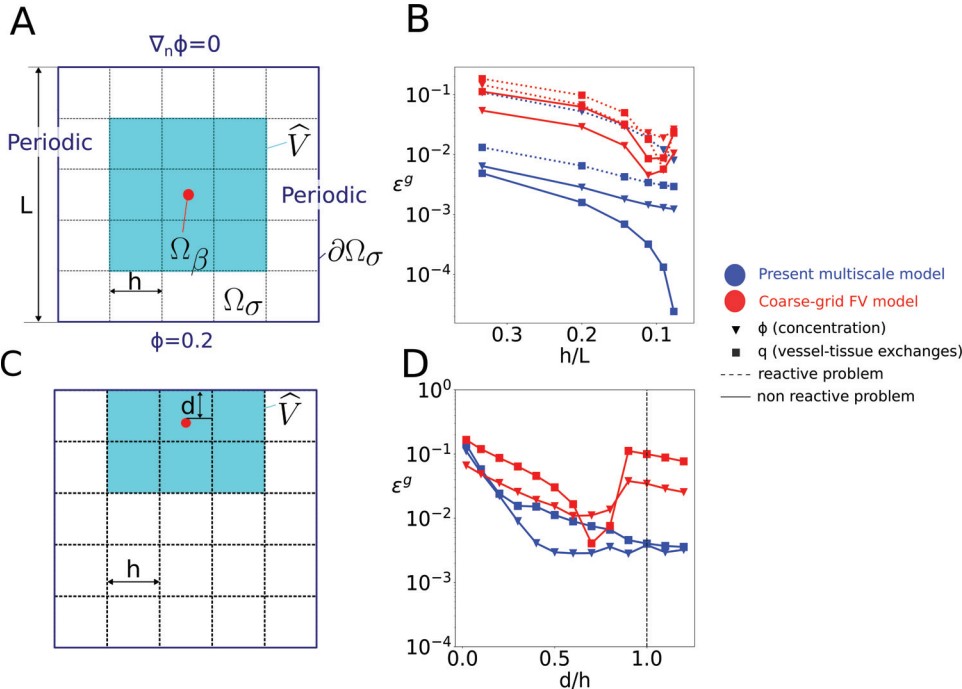

**Fig 4. Error estimation for a single source: Impact of discretization and boundary effects.** A: schematics of the configuration under study, highlighting the detail of the boundary conditions. The domain size is $L = 240\mu$m, the source radius is $R = 4.8\mu$m, and the neighbourhood size is $(30R)^2$, i.e., $n = 3$ for a 5x5 grid ($h/L$=0.2); B: evolution of global errors as a function of grid size for the linear and non-linear problems, and for both the multiscale and the coarse-grid FV model (see legend); C: schematics of the boundary test, for which we use a mesh size $h/L = 0.2$, i.e., a 5x5 grid; D: evolution of global errors as a function of $d$, from $d = 0$ where the source is in contact with the no-flux boundary, to $d = 1.2h$ where the source lies in the contiguous grid-cell. The dashed vertical line illustrates the limit of the boundary cell.

## 3.1 Single source

In this Section, we focus on single source configurations, where we first assess the dependence of numerical errors on mesh size, in the case of coarse meshes (*Case 1.1*). We thus consider grid-cell sizes ($h$) varying from 20$\mu$m to 80$\mu$m, i.e. larger than the source radius and not so small compared to the domain side. In this case, we don't need to consider potentials arising from other sources (Eq 26), thus drawing emphasis away from the size $n$ of the neighborhood $\widehat{V}$ since its purpose is to control the cross influence among sources. We therefore opt for an approximately constant size of $\widehat{V}$ relative to the radius of the source $R$, fixed to 30$R$. This corresponds to $n = 3$ in a 5x5 grid, as displayed in Fig 4A. The exact size of $\widehat{V}$ may slightly vary according to the discretization size $h$ used, as $\widehat{V}$ consists of a discrete number of grid-cells.

Fig 4B illustrates the error evolution with respect to grid-cell size $h$ for a single source, located at the center of the computational domain, and for a combination of boundary conditions (Dirichlet, Neuman, Periodic), as displayed in Fig 4A. Our multiscale model demonstrates remarkable accuracy, achieving global errors below 1% for both flux ($q$) and concentration ($\phi$) estimates even with the coarser grids. Furthermore, these errors are about one order of magnitude smaller than those of the coarse-grid FV approach, since the later lacks a coupling scheme to bridge the scale gap between the source and the coarse-grid. Moreover, the multiscale model errors decrease monotonously with decreasing grid size. In

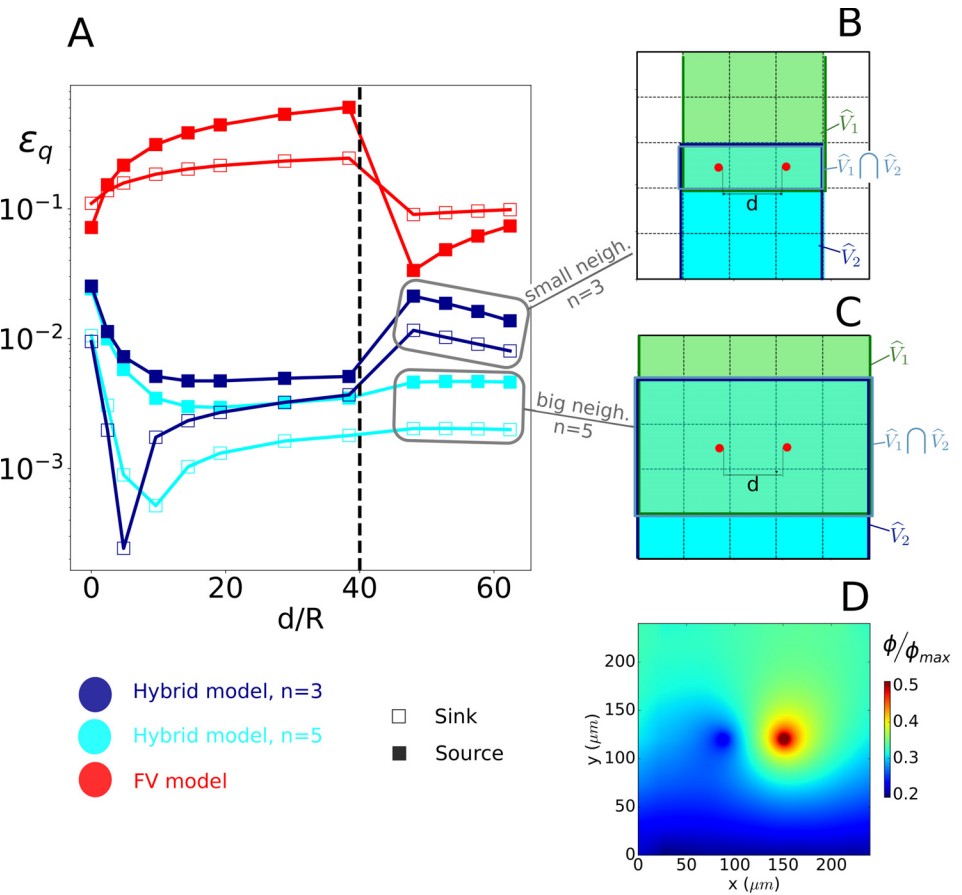

**Fig 5. Error estimation for a single dipole: Interplay between source/sink separation distance and neighborhood size.** A: evolution of the local errors on vessel-tissue exchanges for the source (filled symbols) and for the sink (empty symbols) as a function of distance $d$ between source and sink; B: schematics of the smaller-neighborhood configuration ($n = 3$); C: schematics of the large-neighborhood configuration ($n = 5$); D: reconstruction of the sub-grid concentration field for the case $n = 3$ and $d = 60 \cdot R$. The value of the concentration ($\phi$) is non-dimensionalized by the value of the intravascular concentration in the source. The dashed vertical line in Panel A illustrates the transition between a situation where, for $n = 3$, the intersection of the source and sink neighborhoods contains both of them to a situation where the source and sink lie outside each other's neighborhood.

contrast, the coarse-grid FV approach displays a minimum for grid-cells sizes of about $5R$, as expected from the Peaceman well model [39]. This model bridges the scale gap between the source and the coarse-grid scale as commonly done in geosciences, by relating the value of the scalar field inside the source to the grid via the following flux relationship:

$$q = \frac{K_{eff}(C_v - \tilde{\phi}_k)}{1 + \frac{K_{eff}}{2\pi D}\ln(\frac{R}{0.2h})}$$

When the radius of the source is a fifth of the side length of the grid-cell, the denominator in the above equation is equal to one, and the FV solution (Eq 45) provides the same solution as the Peaceman well model. This occurs at the local minimum observed in Fig 4B, i.e. at approximately $h/L = 0.1$. The coarse-grid FV approach still exhibits errors between $10^{-2}$ and $10^{-1}$ for the smallest grid size considered in this study ($h \sim 4R$).

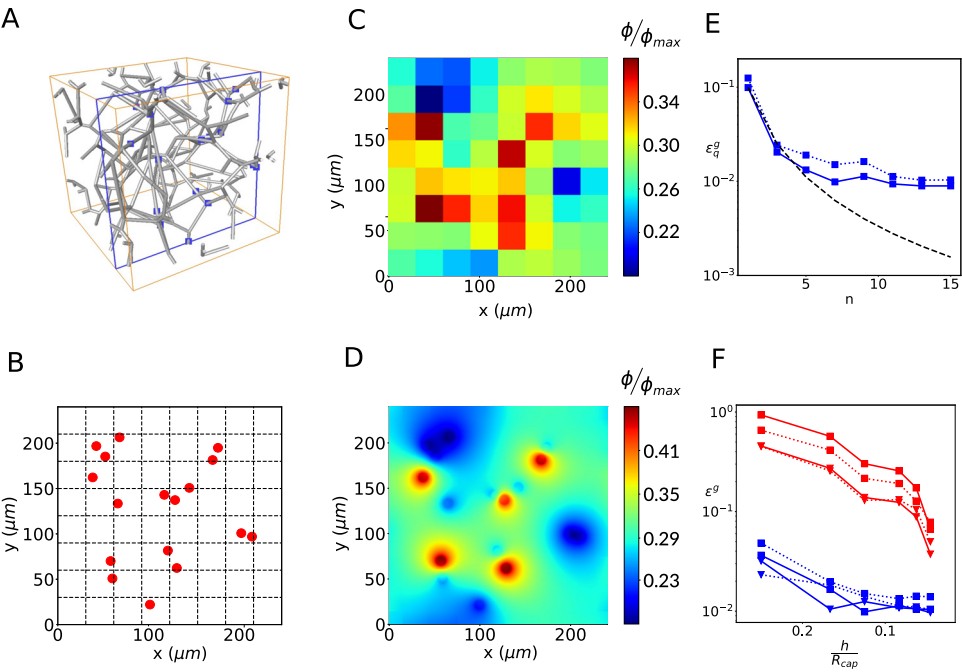

**Fig 6. Error estimations for a realistic source distribution.** A: synthetic capillary bed generated by the method of [60] with cutting plane highlighted in blue; B: intersections of each capillary vessel in A with the cutting plane (red dots) and coarse cartesian grid (dashed lines); C: coarse-grid solution for the concentration field $\tilde{\phi}$ with metabolic consumption; D: sub-grid reconstruction of $\phi$ using $\mathscr{I}_{\phi}$ from Section 2.3.3. E: global errors on vessel-tissue exchanges estimations for a grid size $h = L/16$, therefore 256 FV cells in total. The curve $\frac{\kappa}{(n-1)^2}$ with $\kappa = 0.1$ is represented by the dashed black line; F: evolution of the errors in the estimation of the vessel-tissue exchanges and the concentration field with decreasing mesh cell size and with $n$ chosen so that the size of $\widehat{V}$ is approximately $3L/5 = 30R$.

In contrast, a good balance between mesh-size and accuracy is achieved by the multiscale approach for the 5x5 grid ($h/L = 0.2$) with $n = 3$ (see Fig 4A), with errors on fluxes below 1% for both the linear and non-linear models (see Fig 4B). These parameters will thus be used next except as stated otherwise.

Because the source is located at the center of grid-cell $V_k$, the discrete value of the slow term in this grid cell $\tilde{\jmath}_k$ approximates well the local value of the slow term at source center $\jmath(\mathbf{x}_k)$, so that the concentration $\tilde{\phi}_k = r(\mathbf{x}_k) + \tilde{\jmath}_k$ directly enables the vessel-tissue exchanges to be evaluated using Eq 27. However, this introduces inaccuracies when the source moves away from a grid-cell center, as illustrated in Fig A in S1 Figures, with errors $\varepsilon_q^g$ up to 2.1% when the source is lying on a grid-cell corner. The interpolation scheme introduced in Section 2.3.3 reduces this errors to under 0.3% (Fig A in S1 Figures).

We now worsen the deviation from *Assumption 1* by reducing the distance $d$ between the source and the no-flux boundary (*Cases 1.2* and *2.2*), as illustrated in Fig 4C. Errors reach up to $\approx$10% when the source is in contact with the zero-flux boundary condition ($d = 0$), see Fig 4D. They decrease rapidly with increasing $d$, with $\varepsilon^q < 2\%$ as soon as there is half a grid-cell distance to the boundary. In contrast, the FV solution errors stay consistently around 10% even when the source belongs to a non-boundary grid-cell ($d/h > 1$), except for a minimum for $d/h \sim 0.7$. Similar to the Peaceman well model [39], the local minimum is likely obtained when the logarithmic decrease of the source potential is close to the discrete approximation of its gradient from values at the FV cell's center.

Overall, the single source test-cases highlight how coupling the analytical rapid term to the coarse-grid FV discretization of the slow term improves the numerical resolution of oxygen transport and metabolism within the tissue space. Importantly, these test-cases have been designed to push the limits of the corresponding underlying assumptions, by choosing small computational domains. Given the results shown in Fig 4, we expect to rarely find ourselves in conditions where $\varepsilon \geq 1\%$.

## 3.2 Single dipole

We now test the performance of the multiscale model for a single dipole, i.e., a single source ($\langle C_v \rangle_2 / \phi_{max} = 1$) and sink ($\langle C_v \rangle_1 = 0$). When these are placed close together in the same FV cell, we find ourselves in *Case 2.1*, and *Assumption 2* in Section 2.6 breaks down. In this case, models that don't integrate an analytical description of interactions among sources [37, 72] fail to capture the source to sink interactions. With increasing separation distance $d$ between the source and sink, *Assumption 2* is recovered but, depending on the size of $\widehat{V}$, the deviation from *Assumption 1* may increase (*Case 1.3*). Thus, the dipole situation focuses on the interplay between source separation distance $d$ and neighborhood size $n$ and enables to compare the behavior of the model when the source and sink respective neighbourhoods overlap.

The local errors (Eqs 41 and 42) on the vessel-tissue exchanges are shown on Fig 5A as a function of the separation distance $d$, for the two neighborhood sizes presented in Panel B ($n=3$) and C ($n = 5$). In Panel D, we also show the reconstruction of the concentration field for $n = 3$ and $d = 60R$ using the interpolation function $\mathscr{I}_\phi$. This reconstruction closely approaches the FE reference solution obtained for a dense mesh of over 2, 000 grid cells (Fig B in S1 Figures), i.e, about 100 times the number of cells (5x5) needed to solve for the coarse-grid solution.

When the source and sink both lie in the same grid-cell, i.e., when $d/R$ is below 40, the behavior of these local errors becomes similar whatever the neighbourhood size, since the cross-influence between their potentials is then calculated analytically by the rapid term. When there is no overlap between the two neighbourhoods (e.g. in Fig 5B and for $d/R$ above 40 in the small neighborhood case, dark blue lines in Fig 5A), the errors increase significantly, reaching the upper-bound estimate of errors induced by localized formulation of the rapid term (Section 2.3.2), as evaluated by Eq 36. In contrast, for a larger neighborhood size (light blue lines in Fig 5A and 5C), the errors quickly reach a plateau for an increasing separation distance $d$, consistent with the error obtained for a single source with similar discretization ($h/L = 0.2$ in Fig 4B). This underpins the residual error as the result of the coarse-grid resolution of the slow term and not of potential localization.

## 3.3 Multiple sources

We have shown how errors primarily build up when a source lies in the vicinity of a no-flux boundary (Fig 4B), and in lesser extent when two sources lie close to each other (Fig 5A), respectively. Both situations may arise frequently within the cortex, e.g., close to vessel bifurcations, where three vessel are connected in a single point.

Here, we thus consider a more realistic distribution of sources, obtained using a synthetic network that reproduces the structural and functional properties of cortical capillary beds, following [60] (Fig 6A). Briefly, we take a cross-section of such a network and map its intersections with each vessel (Fig 6A). We thus obtain a realistic map of source distribution, for which $S = 17$ (Fig 6B). We randomly assign one third of vessels to be sources and two third of vessels to be sinks, with periodic boundary conditions.

In Fig 6C and 6D, we show the coarse-grid concentration field and its reconstruction, respectively, for a 8x8 grid ($h/L$=0.125 and $h/R_{cap}$ = 6.25) and $n$ = 5. These clearly show that the model formulation enables enforcing the periodic boundary conditions for the reconstructed, highly-resolved, concentration field, as efficiently as the reference FE approach (Fig B in S1 Figures), even if periodicity at the boundaries doesn't propagate to the scale of the coarse-grid. Furthermore, the evolution of errors with neighbourhood size $n$ (Fig 6E), follows the $1/n^2$ scaling predicted by Eq 36, up to $n$ = 5. For larger values of $n$, a plateau is reached, the value of which ($\sim$1%) corresponds to the residual error associated to deviations from *Assumption 2*, as shown in Sections 3.1 and 3.2. As a result, neither considering finer grid-cells nor increasing the neighborhood size $n$ further reduce this residual errors (see Fig 6E and 6F, respectively).

Furthermore, we note that the numerical errors are only marginally affected when oxygen consumption is taken into consideration (dashed lines in Fig 6E and 6F), showing the robustness of our approach.

In contrast, errors corresponding to the coarse-grid FV model lie consistently one order of magnitude above than the one resulting from the multiscale approach.

## 4 Results: Periarteriolar oxygen concentration gradients

Now that we have shown the ability of our model to efficiently solve for the oxygen concentration field, including around vessels where gradients are the strongest, we turn to its exploitation in the context of brain metabolism. We specifically ask if variations of the radial periarteriolar concentration profiles that were recently measured across cortical layers in awake mice [10] could result from the layer-specific (laminar) increase of capillary density with cortical depth rather than from variations of baseline oxygen consumption.

For that purpose, we consider the typical case of a single penetrating arteriole (PA) and its surrounding tissue, as illustrated in Fig 7A. To account for the capillary-free space that encircles the PA, we include a cylindrical tissue region devoid of capillaries, with typical radius of 100 $\mu$m [10, 24]. Further away, we generate a random spatial distribution of sources with densities approximately matching the capillary density in cortical layer II (Table 2). We deduce the equivalent 2D source density (E2DSD) by using synthetic capillary networks similar to Section 3.3. We then create a randomized but statistically homogeneous distribution of sources following [60, 73]. We assign concentrations at the outer walls of the PA and capillaries, by using an asymptotically large value of $K_{eff}$ and following experimental measurements in layer II [8]. For the capillaries, we assign a random distribution of normalized concentrations at capillary walls $\partial\Omega_{\beta,j}$, drawn from a Gaussian distribution with mean $\phi_{cap} = 0.45\phi_{PA}$ and standard deviation $\sigma = 0.1\phi_{PA}$, approximately corresponding to experimental measurements in layer II (Table 2). We also impose periodic boundary conditions on the limit of the domain to mimic the larger cortical space. Finally, the maximum metabolic rate of oxygen is chosen to be $M = 2.4\mu\text{mol} \cdot \text{cm}^{-3} \cdot \text{min}^{-1}$ (Table 1), an intermediate value within the physiological range. Other transport parameters used to solve the non-linear problem (Eq 39 and 40) are deduced from reference values in the literature (Table 1).

A typical realization of the resulting coarse-grid concentration field, converted to partial pressure ($PO_2$) using the solubility of oxygen in brain tissue [10, 30, 37], is displayed in Fig 7B for a 20x20 grid ($h \sim 20\mu$m). This matches the experimental sampling used in [10] and results in a very good qualitative agreement with the field measured in a 100$\mu$m-deep plane perpendicular to a PA (see Fig 7C, data acquired following [10]). This field can be used to deduce the sub-grid concentration dynamics by interpolation (Eq 30, see e.g. Figs C and D in S1 Figures), as well as the local cerebral metabolic rate of oxygen (CMRO$_2$, see Eq 40 and Fig 7D).

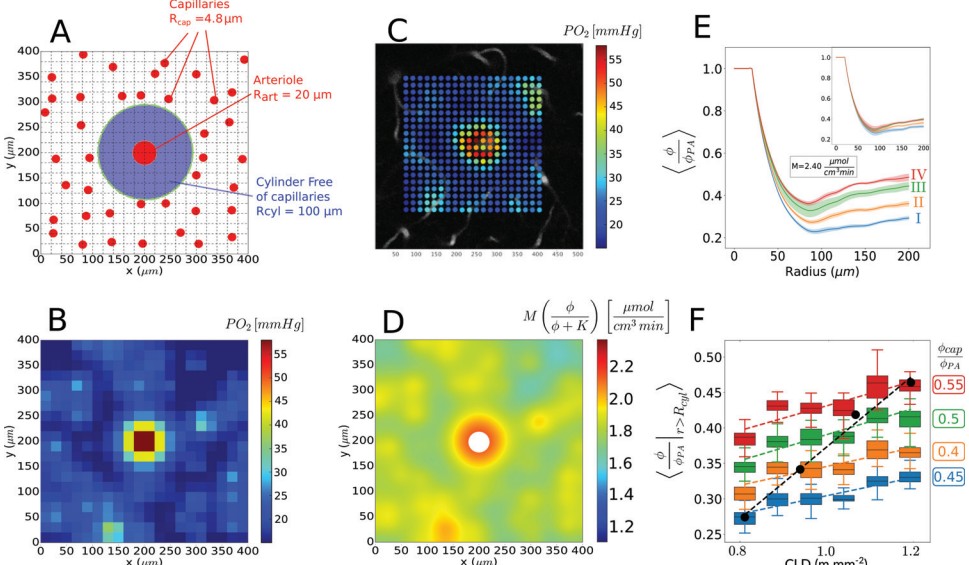

**Fig 7. Effect of capillary density and intravascular concentration on radial periarteriolar concentration profiles.**
A: sketch of simulated configuration, with a random and homogeneous capillary bed for $r > R_{cyl}$ and a capillary-free region around the central PA. The cartesian grid of size 20x20 matches that of experimental sampling used in [10]. B: corresponding coarse-grid partial pressure deduced by linear transformation from the concentration field using the solubility of oxygen in brain tissue [10, 30, 37]. Capillary density and concentration correspond to layer II (Table 2) and $M = 2.4\mu$mol $\cdot$ cm$^{-3}$ $\cdot$ min$^{-1}$. Note that all simulations (panels B, D, E, F) use the same value of $M$ and $n = 10$; C: example of an experimentally sampled oxygen partial pressure field around a PA at $100\mu m$ under cortical surface, i.e. at the interface between layer I and II; D: estimated metabolic consumption deduced from Panel B; E: radial concentration profiles predicted in layers I to IV, each obtained by averaging the results of 30 simulations; Inset: result obtained when only variations of the capillary density are considered; F: resulting spatial average of the tissue concentration for $r > R_{cyl}$, as a function of capillary density for four values of the average capillary intravascular concentration ($\phi_{cap}/\phi_{PA}$ from 0.4 to 0.55) corresponding to the four layers in Table 2; the black dots represent the resulting spatial average of the tissue concentration for $r > R_{cyl}$ for the four different layers, i.e., varying both capillary density and concentration. The black dashed line represents the associated linear fit $(\langle \frac{\phi}{\phi_{PA}}|_{r>R_{cyl}} \rangle = 1.12 \cdot 10^{-3} \cdot$ CLD $- 0.118)$.

Interestingly, the cerebral metabolic rate of oxygen exhibits $\sim \pm 10\%$ variations in the outer region (around capillaries) and up to $\sim \pm 20\%$ in the periarteriolar region, in contrast to the common assumption of a spatial homogeneity [10, 23, 24, 74].

Moreover, the above results can be post-processed to deduce the azimuthal average of the normalized concentration around the PA, as displayed in Fig 7E as a function of the radial distance $r$ to the PA center. In this figure, the plain orange line corresponds to the mean over 30 realizations of source distributions for layer II, while the faint orange areas shows the associated standard deviation. This radial concentration dynamics exhibits three regimes (Fig 7E): 1/

**Table 2. Layer-specific (laminar) variations of capillary density and average intravascular capillary $PO_2$.** The capillary length density (CLD) and depth of the corresponding layers are approximated from data in [20]. The equivalent two-dimensional source density (E2DSD) is deduced using synthetic capillary networks from [60] (Fig 6A). The ratio between arteriole and capillary concentration is approximated from data in [9].

| Layer | CLD [m mm$^{-3}$] | E2DSD [mm$^{-2}$] | Depth [$\mu m$] | $PO_{2,PA}$ [mmHg] | $PO_{2,cap}$ [mmHg] | $\phi_{cap}/\phi_{PA}$ |
|---|---|---|---|---|---|---|
| I | 0.8 | 250 | [0–100] | 99 (= $\phi_{max}/\alpha$) | 39 | 0.4 |
| II | 0.94 | 325 | [100–200] | 92 | 42 | 0.45 |
| III | 1.08 | 400 | [200–300] | 87 | 44 | 0.5 |
| IV | 1.2 | 475 | [300–400] | 85 | 47 | 0.55 |

a constant value for $r \leq R_{\mathrm{PA}}$, i.e. within the PA, consistent with the source potential (Eq 24); 2/ a fast decrease for $R_{\mathrm{PA}} \leq r \leq \sim 0.8 R_{\mathrm{cyl}}$ corresponding to the inner part of the region devoid of capillaries around the PA (see Fig 7A) and 3/ a re-increase followed by a slowly-varying region for larger values of $r$. The presence of a local minima, which can also be observed in the measurements (Fig 7C and Fig E in S1 Figures, dashed lines) suggests that the outer region of the capillary-free cylinder is both fed by the PA and the capillary bed.

Next, as the capillary density approximately increases linearly with depth in the cortex from layer I to layer IV [20], we increased the source density from 250 to 475 mm$^{-2}$ (Table 2). This results in an increase in size of regions with high oxygen concentration around capillaries (see panel B vs. A in Fig C in S1 Figures) and therefore 1/ in a slight decrease of the steepness of radial periarteriolar $PO_2$ gradients averaged over 30 realizations and 2/ in a slight increase of the partial pressure in the plateau region ($r \geq R_{\mathrm{cyl}}$), see inset in Fig 7E. This increase can be quantified by plotting the spatial average $\langle PO_2 / PO_{2,Art} \rangle_{r \geq r_{Cyl}}$ (see isocolor variations in Fig 7F).

Increases of the average intravascular $PO_2$ within the capillary bed (Table 2), which can be speculated based on depth-resolved experimental measurements of vascular oxygen within the cortex [9], result in a higher increase in size of regions with high oxygen concentration around capillaries (see panel C vs. A in Fig C in S1 Figures) and thus to higher increase of $\langle PO_2 / PO_{2,Art} \rangle_{r \geq R_{Cyl}}$ (i.e. black dashed line vs. colored dashed lines in Fig 7F).

Combined together, the increase in capillary density and intravascular $PO_2$ that has been reported experimentally from layer I to layer IV in the cortex of living rodents leads to an even faster decrease of the perivascular concentration gradient (Fig 7E). This results in a faster increase of $\langle \phi / \phi_{2,PA} \rangle_{r \geq R_{Cyl}}$ from layer I to layer IV, see black dots in Fig 7F. For a constant value of $M$, this yields an increasing metabolic rate of oxygen from layer I to layer IV (panel D vs. A in Fig C in S1 Figures), consistent with the increased density of mitochondrial cytochrome oxidase through these layers [10, 75, 76].

Noteworthy, similar results have been obtained for different values of the maximal metabolic rate of oxygen $M$ within the physiological range (Table 1), as illustrated in Fig D in S1 Figures. The only notable difference is that the amplitude of the dip in the radial concentration profile decreases with decreasing values of $M$ (Fig E in S1 Figures). For the set of parameters representative of layers I and II, this leads to a monotonous decrease of the average radial concentration followed by a region with nearly constant oxygen concentration, similar to experimental measurements reported in [10, 24], as soon as $M \leq 1.2 \mu\mathrm{mol} \cdot \mathrm{cm}^{-3} \cdot \mathrm{min}^{-1}$ (Fig E in S1 Figures). As a result, the oxygen dynamics in the close vicinity of the arteriole ($r < R_{cyl}/2$) may be highly similar in different cortical layers for different values of $M$ (e.g. $M = 2.4 \mu\mathrm{mol} \cdot \mathrm{cm}^{-3} \cdot \mathrm{min}^{-1}$ in layer IV, see red line in Fig 7E vs. $M = 1.6 \mu\mathrm{mol} \cdot \mathrm{cm}^{-3} \cdot \mathrm{min}^{-1}$ in layer I, see blue dashed-dotted line in Fig E in S1 Figures).

Altogether, the present model suggests that laminar variations of the capillary density may be sufficient to explain the differences in periarteriolar radial oxygen profiles measured at different depths within the cortex [10], without any variation of the maximal cerebral metabolic rate of oxygen $M$. If laminar variations of intravascular capillary $PO_2$ are also considered, the predicted differences are even larger than the experimental ones. This demonstrates the interplay between metabolic consumption, capillary density and intravascular availability of oxygen in the capillary bed to determine the radial oxygen gradient in the vicinity of PAs. This makes it difficult to consider the steepness of the radial periarteriolar oxygen profile as a surrogate for the baseline oxygen consumption, with the potential to reconcile recent experimental measurements with the idea that laminar variations of capillary density could reveal underlying differences in metabolic load.

## 5 Discussion

In this paper, we revisited the problem of oxygen transport and metabolism in the brain parenchyma, with the goal to introduce a simple, scalable and accurate numerical method for its direct resolution. Getting inspiration from previous work on blood flow and oxygen transport in the brain [22, 30, 45, 64] and on mixed-dimensional problems in applied mathematics for geosciences [39, 56, 57] and biology [48], we applied the notion of operator-splitting, which allowed us to describe the oxygen concentration field in the parenchyma as the sum of a slow and a fast varying contributions. The slow contribution was treated using a classic finite volume approach on a coarse grid, while the fast contribution was described using Green's functions that allowed to analytically capture the sub-grid perivascular concentration gradients. This made it possible to locally bridge the scale gap between sources and the coarse-grid with higher flexibility than [39, 41] regarding the position of the sources within the coarse grid-cells, including multiple sources within a single grid-cell, the proximity of the boundaries, and the control of the matrix sparsity thanks to the the size of the neighbourhood ($\hat{V}$). Similarly to singularity removal approaches [45, 48, 57], this also made it possible to mix the dimensionality reduction of the Green's functions approaches [30, 32, 51] with the versatility of FV methods [56, 57, 72]. Moreover, solving for a slowly varying background concentration field, thanks to a change of variable, offers a huge advantage with respect to the Green's functions resolution since it allows for a localization of the source potentials, thereby providing a much sparser system. In addition, this enable the use of a much coarser mesh that reduces considerably the size of the system compared to FV or FE methods, but without loss of precision thanks to the sub-grid reconstruction of the concentration field (Figs 5D and 6D). This, in turn allows the addition of non linear volume terms (metabolism) without significant loss of accuracy (Figs 4B, 4D and 6E).

To provide rigorous but still intelligible mathematical developments, we focused on a two-dimensional version of the problem (Section 2). In this way, we were able to introduce the localization scheme enabling us to control the bandwidth of the associated linear system of equations, by manipulating the size of the region over which the interactions with nearby vessels are accounted for analytically (scalability). We demonstrated the existence of an optimal size for this region, above which the errors induced by localization are smaller than those induced by deviations from local azimuthal symmetry of the concentration field around each vessel (see Fig 6E). This emphasizes the importance of comparing the results of any simplified model for oxygen transport in the brain parenchyma with a reference solution that is able to fully resolve these deviations. This is neither the case if only single vessels with Krogh-type configurations are considered for validation, as in [10, 23, 27, 37], or if the discrete version of the problem is compared to the corresponding continuous version (i.e., comparing the solution of Eq 33 to the solution of Eq 6 instead of Eq 1), as in [58, 66, 72, 77]. To our knowledge, such comparisons had never been performed before in this context. Crucially, they enabled to provide careful estimates of the numerical errors associated to the use of coarse meshes, demonstrating the unprecedented balance between reduction of problem size and minimization of errors associated to our method, compared to previous strategies in the literature (accuracy). With this regard, it is worth insisting that we designed test cases that enabled the origins of errors to be understood by purposefully choosing configurations with deviations from the model underlying assumptions (Section 3). Thus, all errors provide upper-bounds of the errors expected when considering larger, physiological-like problems. Moreover, the mathematical groundwork provided by the Green's function framework (Section B in S1 Methods) permitted to trace back the source of errors to specific modeling assumptions, which in turn offers a rationale for choosing the model parameters, including discretization and neighborhood size.

Finally, the method makes use of a cartesian mesh independent of vessel locations, thereby belonging to the class of mesh-less approaches [38]. In contrast with the widespread semi-analytical methods [30, 32, 34, 51, 65] that require computationally intensive Fourier transforms to enforce conventional boundary conditions such as Neumann and Dirichlet [30], it also shows remarkable versatility with regard to the boundary conditions that can be handled (simplicity).

Of course, the two-dimensional version of the problem we considered doesn't enable coupling oxygen transport in the parenchyma with oxygen transport in blood vessels, because intersecting the vascular network with a plane yields disconnected vascular sources, as schematized in Fig 1. Thus, in the present work, intravascular concentrations have been treated as inputs, while in a three-dimensional version they should be treated as unknowns, with additional blocks in the final system accounting for intravascular transport, as highlighted in [45, 52, 66]. Together with previous work by our group that focused on revisiting intravascular transport [22], these will provide the foundations for an extension to three dimensions. The fact that the integrated potential arising from a circular source can be written analytically (Eq 24 and Section B in S1 Methods) is a peculiar characteristic of the 2D model. In 3D, additional errors may arise due to the approximations needed to estimate the potential of a small cylindrical element used to discretize the vascular networks, that will require careful evaluation in the spirit of [78, 79].

However, considering two-dimensional situations already offers the opportunity to investigate important physiological questions, as illustrated in Section 4. Thanks to the computational efficiency of our method, we were able to easily generate the large number of synthetic data (600 simulations per layer and 30 measurements per simulation) needed to incorporate statistical information about the capillary bed available in the literature [9, 20]. By this way, we shed new light on the interpretation of experimentally measured variations of periarteriolar oxygen profiles [10] in the context of laminar variations of capillary density and their relationships with baseline cerebral metabolic load. Due to the non-local nature of blood flow, understanding the measured variations of average intravascular concentration with depth will require a fully coupled three-dimensional analysis of oxygen exchange in the brain. In this regard, it is worth noting that the matrix assembly process only depends on the location of vessels in the considered domain and of the size of grid-cells and neighbourhood $\widehat{V}$, and requires to be performed only once when the structure of the vascular network is known. Besides parametric analyses, this will pave the way for the inverse modelling of brain metabolism from three-dimensional oxygen measurements. Inverse modelling indeed requires fast and precise forward model resolutions, overcoming the geometric simplifications of the Krogh cylinder type, which are at the basis of all current work that aims at measuring the cerebral metabolic rate of oxygen [10, 24, 26].

## 6 Conclusion

We developed a multi-scale model describing the spatial dynamics of oxygen transport in the brain that is simple, conservative, accurate and scalable. Our strategy was to consider that the oxygen concentration in the parenchyma is the result of a balance between contributions at local (cell metabolism, delivery by neighbouring arteriole) and larger (capillary bed) spatial scales. This allowed us to split the oxygen concentration field into a slow and a fast varying terms, underlying the separation of scales between local and distant contributions. Doing so allowed us to combine a coarse-grid approach for the slow-varying term to a Green's function approach for the fast-varying terms. This resulted in a computationally efficient model that was able to capture precisely gradients of concentration around microvessels and to describe boundary condition with flexibility (Dirichlet, Neumann, and periodic) along with the non-linear metabolic activity by the cells.

We then compared our model with reference solutions of the oxygen transport problem in scenarios of increasing complexity. We showed that our model was able to maintain small errors, even in scenario where the separation of scales was challenged or when azimuthal variations of flux were no longer negligible, demonstrating the robustness of the model.

While the present multi-scale model focused on two-dimensional problems, it has been designed to be easily extended to three-dimensional problems by adapting the expression for the source potential and by including an intravascular description of oxygen transport, both of which being available in the literature (see e.g. [22, 30, 45]).

Despite this limitation, we showed that the model was already capable of generating synthetic data reproducing the heterogeneous distribution of oxygen in the brain parenchyma. Doing so, we showed that periarteriolar gradients were the result of the balance between local cellular oxygen consumptionand supply by not only the neighbouring arteriole but also distant capillaries, thus reconciling recent measurements of periarteriolar oxygen gradients across cortical layers with the fundamental idea that variations of vascular density within the depth of the cortex may reveal underlying differences in neuronal organization.

## Supporting information

**S1 Figures. Contains supplementary figures A to E.**
(PDF)

**S1 Methods. Contains supplementary methods Sections A to E.**
(PDF)

## Acknowledgments

We gratefully acknowledge Maxime Pigou for technical help with the development and maintenance of the code and Franca Schmid for carefully reading our manuscript.

## Author Contributions

**Conceptualization:** David Pastor-Alonso, Franck Boyer, Michel Quintard, Yohan Davit, Sylvie Lorthois.

**Data curation:** David Pastor-Alonso.

**Formal analysis:** David Pastor-Alonso, Michel Quintard, Yohan Davit, Sylvie Lorthois.

**Funding acquisition:** Sylvie Lorthois.

**Investigation:** David Pastor-Alonso, Natalie Fomin-Thunemann, Yohan Davit, Sylvie Lorthois.

**Methodology:** David Pastor-Alonso, Maxime Berg, Franck Boyer, Yohan Davit, Sylvie Lorthois.

**Project administration:** Sylvie Lorthois.

**Resources:** Sylvie Lorthois.

**Software:** David Pastor-Alonso.

**Supervision:** Yohan Davit, Sylvie Lorthois.

**Validation:** David Pastor-Alonso.

**Visualization:** David Pastor-Alonso.

**Writing – original draft:** David Pastor-Alonso, Sylvie Lorthois.

**Writing – review & editing:** David Pastor-Alonso, Maxime Berg, Franck Boyer, Natalie Fomin-Thunemann, Michel Quintard, Yohan Davit, Sylvie Lorthois.

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
