## [Decision Letter · Decision Letter 0]

1 Dec 2023

Dear Dr. Lorthois,

Thank you very much for submitting your manuscript "Modeling oxygen transport in the brain: An efficient coarse-grid approach to capture perivascular gradients in the parenchyma" for consideration at PLOS Computational Biology.

As with all papers reviewed by the journal, your manuscript was reviewed by members of the editorial board and by several independent reviewers. In light of the reviews (below this email), we would like to invite the resubmission of a significantly-revised version that takes into account the reviewers' comments.

We cannot make any decision about publication until we have seen the revised manuscript and your response to the reviewers' comments. Your revised manuscript is also likely to be sent to reviewers for further evaluation.

Sincerely,

Daniel A Beard

Section Editor

PLOS Computational Biology

Daniel Beard

Section Editor

PLOS Computational Biology

Reviewer's Responses to Questions

**Comments to the Authors:**

Reviewer #1: The distribution of oxygen levels in brain and other tissues depends on the structure and flow distribution of the microvascular networks that supply the tissues. Oxygen transport to tissue is achieved by convection in blood vessels and diffusion from vessels to surrounding tissue. Theoretical models provide an approach for analyzing oxygen transport, such as deducing local oxygen consumption rates from experimental measurements of oxygen levels. Here the authors focus on methods for modeling oxygen diffusion in tissue, taking into account that the oxygen level at a point depends on oxygen supply by multiple nearby vessels, not just a single capillary as in the classic Krogh cylinder model. A challenge for such models is that if conventional finite-difference or finite-element methods are used, fine meshes are required to resolve the steep gradients around each vessel, resulting in very large systems of unknowns, especially in 3D. This problem can be mitigated by using a Green’s function approach, where the oxygen field is represented by a superposition of point-source solutions to the Laplace equation. While this reduces the number of unknowns, it leads to a problem in which the oxygen field at any point depends on all the unknown source strengths. In computational terms, a large linear system must be solved, where all the matrix entries are non-zero.

Here, the authors present a method in which the domain is split into small regions, and the solution in each region is represented by the sum of a “slow” solution that has a simple variation in each region and a “fast” solution that represents the rapidly varying solution due to a source term. The point-source solution of the Laplacian is represented explicitly only on the small region containing the source and a relatively small number of surrounding regions. Beyond that distance, its effects are represented via the “slow” solutions. This approach is adapted from methods used in geoscience reservoir simulations. While this approach adds considerable complexity, it has the key advantage of resulting in a sparse linear system, so that larger systems can in principle be solved before running into computational limits. This approach represents a novel and useful contribution. The application to periarteriolar oxygen gradients is appropriate and relevant to recent published work on estimation of oxygen consumption rates. The work is thoroughly and carefully presented, although some key points are easily overlooked in the large amount of detail provided.

Specific comments

1. Line 72: The limitations of standard numerical methods (finite difference, finite element, etc.) are well addressed in this section. However, the capabilities and limitations of the well-established Green’s-function approach should also be discussed in this section. (See for example Celaya-Alcala JT et al., Simulation of oxygen transport and estimation of tissue perfusion in extensive microvascular networks: Application to cerebral cortex. J Cereb Blood Flow Metab 41: 656-669, 2021.) The key point in this context is that the Green’s-function approach requires solving a large non-sparse system.

2. Line 201: “for harmonic functions such as s(x), from Gauss’s harmonic function theorem [54], …, this average value is equal to the value at the cell’s center.” This is not true for non-circular domains. For example, consider the function cosh(pi x) cos(pi y) on the unit square centered on the origin. The average on the domain is 4 sinh(pi/2)/pi^2, which is 0.932, but the value at the center is 1. At best, this must be considered as an approximation.

3. Line 273: The presentation of the model is 2D, with prescribed oxygen levels in vessels, and the coupling of diffusive and convective transport is not addressed. It is stated that “The rest of the model presented in section 2.3 can be simply extrapolated to 3D.” However, a proper 3D treatment would require modeling convection in blood vessels, which is not a simple extension.

Minor comments

1. P. 1, abstract last sentence, also line 737, should be “reconciling”, line 617 should be “reconcile”

2. Line 59, “breakthroughs”

3. Line 142, “lineic” is not a word. Say “per unit length”

4. Line 630: “capture”

5. Lines 735 and 738, “periarteriolar”

6. P. 37, ref. 39 is incomplete

Reviewer #2: This is a very detailed and compelling article on an improved modeling algorithm to overcome known problems dealing with varying spatial gradients in heterogenous environments. The application to brain is new and the description of the algorithm is reasonable, at least regarding differences/advancements of the proposed method. The actual implementation of the algorithm is detailed and intricate, and remains difficult to grasp at times. This is not necessarily a problem, especially if the code is provided to sort through nuances beyond the textual description. This is only a minor concern. The main issue with the work submitted is that a big portion of the goal is to solve the inverse problem of determining metabolic rates, this part remains a little weak and unclear in that there is no direct simulation to determine errors based on metabolic rates. This is likely embedded in the sources and sinks portion of the work (a sink presummably being a large metabolically active unit and a source presumably a penetrating artery with high oxygen), but this presentation is not sufficiently clear. Another potential issue is that the work ends without testing for regimes or manipulations where simpler models could do well enough at estimating metabolic rates; for example, if oxygen supply is doubled but M remains the same or if oxygen supply is halved and M remains the same. It would be a novel contribution to describe conditions or manipulations that would help mitigate errors from either standard model vs. this model.

It would be useful to see results at various capillary densities with the same input (flow and oxygen) and same metabolic rate, since different cortical layers have different densities (acknowledged by the authors) but these results are not systemically presented.

Have the authors attempted to model the more superficial layer where the oxygen tensions are relatively high? Especially in Layer 1 where most processes exist vs. Layer 2 or deeper layers? This likely sets up other difficulties but it would be interesting to describe if or how this could be done.

Define variable s and vector n for completeness, especially since n is also used for other purposes. Probably also Card (cardinality).

ln in several instances like Eq 24 is likely natural log and should not be italicized.

**Have the authors made all data and (if applicable) computational code underlying the findings in their manuscript fully available?**

Reviewer #1: Yes

Reviewer #2: **No: **Not at this time but they indicate they will after acceptance.

PLOS authors have the option to publish the peer review history of their article (what does this mean?). If published, this will include your full peer review and any attached files.

Reviewer #1: No

Reviewer #2: No
---

## [Decision Letter · Decision Letter 1]

5 Mar 2024

Dear Dr. Lorthois,

We are pleased to inform you that your manuscript 'Modeling oxygen transport in the brain: an efficient coarse-grid approach to capture perivascular gradients in the parenchyma' has been provisionally accepted for publication in PLOS Computational Biology.

Best regards,

Daniel A Beard

Section Editor

PLOS Computational Biology

Daniel Beard

Section Editor

PLOS Computational Biology

Reviewer's Responses to Questions

**Comments to the Authors:**

Reviewer #1: The authors have responded appropriately to my previous critique.

Reviewer #2: The authors have addressed my concerns and the results in Sup Fig 4 and 5 are very interesting!

**Have the authors made all data and (if applicable) computational code underlying the findings in their manuscript fully available?**

Reviewer #1: Yes

Reviewer #2: Yes

PLOS authors have the option to publish the peer review history of their article (what does this mean?). If published, this will include your full peer review and any attached files.

Reviewer #1: No

Reviewer #2: No

---

## [Editor Report · Acceptance letter]

6 May 2024

PCOMPBIOL-D-23-01591R1 

Modeling oxygen transport in the brain: an efficient coarse-grid approach to capture perivascular gradients in the parenchyma

Dear Dr Lorthois,

I am pleased to inform you that your manuscript has been formally accepted for publication in PLOS Computational Biology. Your manuscript is now with our production department and you will be notified of the publication date in due course.

With kind regards,

Anita Estes
